# Compress & Cache: Vision token compression for efficient generation and retrieval

**Adrian Bulat**[*1,2]    **Yassine Ouali**[*1]    **Georgios Tzimiropoulos**[1,3]

[1]Samsung AI Cambridge    [2]Technical University of Iasi    [3]QMUL

## Abstract

This work aims to compress the vision tokens of an LVLM into a representation that is simultaneously suitable for (a) generative and (b) discriminative tasks, (c) is nearly lossless, and (d) storage-efficient. To this end, we propose C&C, a novel compression method that leverages the LVLM itself for task-agnostic visual token compression. Unlike prior methods that perform token reduction on-the-fly, our approach offloads computation to a dedicated, upfront indexing stage, effectively decoupling compression from generation. This enables learning more powerful representations for generation during inference. At the core of C&C is a "double-forward pass" training strategy. During the first forward pass, the LLM (of the LVLM) creates a bottleneck by compressing the dense visual tokens into a few summary tokens. Subsequently, the second forward pass processes the language instruction(s) alongside the summary tokens, used as a direct replacement for the image ones. The training of C&C is guided by two key losses: an autoregressive loss applied after the second pass that provides a direct optimization objective for reconstructing the original information flow, and a contrastive loss applied after the first pass to bolster the representational strength of the summary tokens, particularly for discriminative tasks. Moreover, we propose stage-specific adapters for further enhancing performance. C&C produces highly informative compressed representations. An in-depth ablation study confirms the efficacy of our approach. For generative tasks, we achieve a $2\times$ higher compression rate without compromising capabilities, setting a new state-of-the-art. For discriminative tasks, we establish new state-of-the-art results on image retrieval and compositionality benchmarks.

## 1   Introduction

Large Vision Language Models (LVLMs) are LLMs [8, 19] that, in addition to text, are capable of integrating and processing visual information as input context [24]. Being able to reason across both vision and language, they are suitable for a wide range of use cases such as image captioning, VQA, and multimodal chatbots. A key bottleneck for their efficient deployment is the large number of input visual tokens, which often dominate the sequence length compared to the language instruction(s). Recent efforts to improve their efficiency have primarily focused on *on-the-fly* token compression [27, 47, 3, 16]. These approaches aim to prune or dynamically condense visual tokens during the online inference process for a given input image and query. While beneficial for single-pass efficiency, these methods operate without a dedicated upfront compression or caching stage and thus limit the capacity of the compressor. Moreover, in the context of retrieval-augmented generation (RAG), they are not aligned with a typical RAG setting, whereby the image and documents are available a priori.

In this work, we explore a different paradigm for LVLM visual token compression that leverages offline processing and caching. Instead of performing token reduction during every inference step, we propose to perform a computationally more intensive compression step once for a given image

---

[*]Denotes equal contribution.

39th Conference on Neural Information Processing Systems (NeurIPS 2025).

to generate a small set of general-purpose summary tokens. These summary tokens are cached and then used directly for subsequent inference queries (*i.e.* online processing, RAG). See Fig. 1. This decoupling of compression and generation allows for a more powerful and versatile representation to be learned during the offline caching phase. Importantly, the representations learned are also suitable for discriminative tasks (*i.e.* retrieval), unlike all prior works, which focus solely on generation.

To this end, we propose Compress and Cache (C&C), a novel token compression approach constructed to support the decoupled indexing (caching) and generation stages. Our core methodological insight and contribution is that the LVLM itself can be adapted to perform the necessary visual compression, leveraging a newly proposed "double-forward pass" training strategy. Specifically, the first forward pass through the LVLM functions as the offline compression phase: trainable summary tokens are processed alongside the image tokens and a predefined prompt guiding the general-purpose visual compression, creating an information bottleneck. The second forward pass simulates the online inference phase: instead of passing the image, the produced summary tokens and the language instruction are fed into the LLM (of the LVLM) for optimization with next-token prediction loss.

A second methodological contribution of ours is to optimize the summary tokens not just for autoregressive generation, but also for discriminative tasks (*e.g.* image-text retrieval). This is achieved through the incorporation of a contrastive loss applied on the summary tokens, after the first forward pass. A significant finding is that this contrastive loss not only enables discriminative capabilities but also proves beneficial for improving the accuracy of the generative tasks.

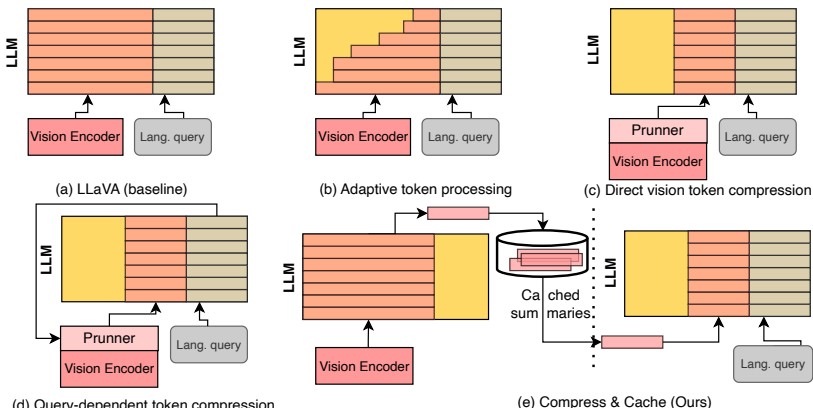

Figure 1: Different paradigms for compressing the visual tokens in LVLMs: (a) LLaVA (baseline) [34]: all vision tokens are used; (b) Adaptive token processing (*e.g.* [4]): the vision tokens are pruned dynamically within each LLM layer. It requires all vision tokens (hence, storage inefficient); (c) Direct vision token compression (*e.g.* [47]): a separate module is learned separately or jointly with the vision encoder; (d) Query-dependent token compression (*e.g.* [27]): the compression depends on the LLM features of each query; (e) Our setting: the LVLM itself produces the compressed representations, trading-off offline processing with superior compression performance.

Overall, **we make the following contributions**:

- We introduce C&C, a novel two-staged method that leverages the LVLM itself for token compression. C&C employs a novel "double-forward pass" training strategy to learn a compact visual representation in the form of condensed summary tokens during an offline phase. The summary tokens are then cached and later used for efficient online inference. C&C effectively disentangles compression from generation, allowing for learning powerful compressed representations.

- We show that the summary tokens learned by C&C are effective for *both* generation and discriminative tasks, such as image-text retrieval. We achieve this dual capability by incorporating a contrastive loss along the autoregressive cross-entropy loss during training. This combined loss is shown to be crucial not only for discrimination but also for enhanced generative performance.

- For generative tasks, C&C achieves a $2\times$ higher compression rate compared to prior methods without compromising capabilities, setting a new state-of-the-art. For discriminative tasks, our method establishes new state-of-the-art results on key image retrieval and compositionality benchmarks. For Visual RAG, we outperform the state-of-the-art VisRAG retriever with a $3.8\times$ smaller model, and almost match the uncompressed baseline for generation, despite using $24\times$ fewer tokens.

## 2 Related work

**Token compression/reduction in LVLMs:** While achieving remarkable multimodal capabilities, LVLMs [35, 34, 56, 65, 55, 2, 30, 7] often face significant computational cost, largely due to the LLM having to process a substantial number of visual tokens (*e.g.* 576 in [34]). To alleviate this, recent research has focused on reducing the number of visual tokens fed as input into the LLM [27, 47, 59, 3, 16]. These works operate under an *on-the-fly* paradigm, performing the reduction during inference (see Fig. 1 for a conceptual comparison). Various strategies have been proposed: PruMerge [47] and [64] use training-free heuristics based on spatial token similarities with the global token or with the text query, while methods like [29] and Matryoshka-style techniques [3, 16, 7] train specific modules (e.g., attention layers or convolutions) to learn fixed compressed or nested representations. Other methods implement dynamic token reduction within the LLM layers [55, 4] or condition the reduction on the language query [27]. [57] adapts the attention pattern of the LVLM to store the visual information as part of a compressed KV representation, incurring high storage costs. QueCC [27] conditions its token selection on an LLM-produced embedding derived from the user query, a design that requires recomputation of the visual token reduction for every new instruction.

The proposed C&C fundamentally differs from these prior works by operating under an offline compression and caching paradigm (rather than on-the-fly token reduction), making it ideal for RAG and on-device deployment. Specifically, C&C performs an upfront compression step to generate a task-agnostic, cached summary representation of the image using a newly proposed "double-forward pass" training strategy that leverages the whole LVLM for compression. C&C offers several key advantages. Firstly, it decouples the compression step from online inference, allowing for a more sophisticated offline compression process. Secondly, C&C's summary tokens are optimized (via a contrastive loss) to support both generative and discriminative tasks. This dual capability is a key distinction. Finally, C&C achieves state-of-the-art results surpassing query-dependent methods like QueCC without incurring the computational cost of recomputing visual tokens for every new query.

**Discriminative LVLMs:** Very recently, a series of works [20, 21, 17] have explored the task of converting LVLMs into discriminative models. For example, [17] directly aligns a pretrained LLM with a pretrained CLIP vision encoder. E5-V [20] through text-only contrastive training converts a generative LVLM into a discriminative one, while [21] expands it to multi-modal retrieval. One major limitation of these works is the loss of generative abilities post-adaptation. In this work, we address this very issue, creating a unified model that excels at both generative and discriminative tasks, surpassing recently proposed LVLM adaptations for image-text retrieval and compositionality.

## 3 Method

### 3.1 Preliminaries

We implement C&C on top of LLaVA-1.5 [34], leaving all architectural components unchanged. The LLaVA model consists of a pretrained CLIP vision encoder $g(.)$, a projection matrix $\mathbf{W}$, and an LLM $f(.)$. The input image $\mathbf{X}_v$ is passed to CLIP to produce vision embeddings $\mathbf{H}_v = g(\mathbf{X}_v)\mathbf{W}$. The language embeddings $\mathbf{H}_q$ are obtained from the input language instruction $\mathbf{X}_q$. Finally, the concatenated vision and language embeddings are passed to the LLM to compute the answer (output) embeddings $\mathbf{H}_a = f(\mathbf{H}_v; \mathbf{H}_q)$, which is decoded to the corresponding answer (output) sequence $\mathbf{X}_a$.

Although autoregressive in nature, recently, it was shown that the model can be run in discriminative mode, producing image-text embeddings for matching *à la* CLIP [20]. Using the prompt $\mathbf{X}_p$ "summarize the above image in one word" (or similar), the image embedding is produced as $\mathbf{e}_v = \mathbf{H}_a[-1]$, $\mathbf{H}_a = f(\mathbf{H_v}; \mathbf{H}_p)$ ($\mathbf{H}_p$ is the language embedding of $\mathbf{X}_p$). Analogously, and given a text query $\mathbf{X}_{query}$, the text embedding is constructed as $\mathbf{e}_t = \mathbf{H}_a[-1]$, $\mathbf{H}_a = f(\mathbf{H}_{query}, \mathbf{H}_p)$ ($\mathbf{H}_{query}$ is the embedding of a $\mathbf{X}_{query}$). The image-text similarity is computed as $s = \cos\_\text{sim}(\mathbf{e}_v, \mathbf{e}_t)$.

### 3.2 Double forward bottleneck algorithm

Assuming a fixed LVLM architecture, its inference cost is defined by the input sequence length, which turns out to be dominated by the length (number) $k$ of the vision embeddings $\mathbf{H_v} \in \mathbb{R}^{k \times d}$. For a LLaVA model, $k = 576$, which is significantly higher than the typical text query length

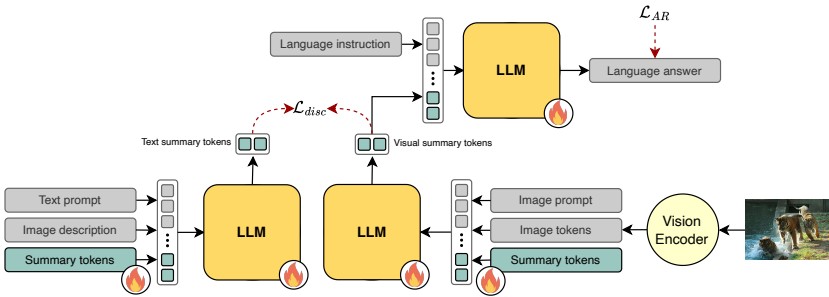

Figure 2: **C&C training pipeline:** A first forward pass from the LLM creates a bottleneck by condensing the visual information into a small number of visual summary tokens. Then, using the same LLM with shared weights, a second forward pass processes the language instruction(s) alongside the summary tokens for training with a next-token prediction loss $\mathcal{L}_{\mathrm{AR}}$ (see Sec. 3.2). A contrastive loss $\mathcal{L}_{\mathrm{disc}}$, applied after the first pass, further boosts the representation strength, especially for discriminative tasks (see Sec. 3.3). Trainable components are marked with 🔥. Note, that different LoRA adapters are used depending on the stage: compression or generation.

and answer [27]. In this work, our goal is to derive a compressed visual token representation $\mathbf{H}_v^c \in \mathbb{R}^{k' \times d}$ where $k' \ll k$ without compromising the model's accuracy. Importantly, besides improving subsequent runs, a small sequence length also opens the path to offline pre-processing, whereby one can pre-compute, cache, and re-use the compressed representation $\mathbf{H}_v^c$ without having to process the original image again.

Departing from all previous approaches for token compression/reduction, we take a totally different path by proposing to leverage the LVLM itself (*i.e.* the LLM of the LVLM) to self-compress the visual tokens. Our motivation for this is multifold. Firstly, LLMs have already excelled at text summarization [63, 36], hence, we propose to utilize them for image summarization (*i.e.* compression). However, as summarization with text, due to quantization (*i.e.*, tokenization), is inefficient with respect to the sequence length, we instead perform this summarization in a continuous latent space. Secondly, the LLM of the LVLM has already been recently utilized to compute discriminant image-text embeddings [20]. However, these embeddings cannot be used for generation. To this end, we propose a "double-forward pass" training strategy whereby visual summary tokens at the output of the model are directly trained to highly compress visual information for both generation and discrimination. See Fig. 2 for an overview.

More specifically, given an input image $\mathbf{X}_v$, we introduce the summary tokens *i.e.* learnable input embeddings $\mathbf{H}_r \in \mathbb{R}^{k' \times d}$ which evolve into the compressed vision embeddings $\mathbf{H}_v^c \in \mathbb{R}^{k' \times d}$ after *a first forward pass from the LLM of the LVLM*:

$$[;,;,\mathbf{H}_v^c] = f(\mathbf{H}_v; \mathbf{H}_p; \mathbf{H}_r), \qquad (1)$$

where $\mathbf{H}_p$ are the embeddings of the prompt $\mathbf{X}_p$ "Summarize the image in a few words.". Clearly, during this pass, $\mathbf{H}_r$ interact with both $\mathbf{H}_v$ and $\mathbf{H}_p$. As the transformed embeddings $\mathbf{H}_v^c$ are query-agnostic, for subsequent instructions/queries, the LLM simply takes as input $\mathbf{H}_v^c$ instead of $\mathbf{H}_v$.

To learn the compressed representation $\mathbf{H}_v^c$, during training, we perform *a second forward pass from the LLM of the LVLM* where this time only $\mathbf{H}_v^c$ and the language instructions/embeddings are passed to the LLM. An autoregressive loss is applied at the output of the second forward pass:

$$\mathcal{L}_{\mathrm{AR}} = -\sum_{i=1}^{L} \log\left(p_\theta(x_i | \mathbf{H}_v^c, \mathbf{X}_{q,<i}, \mathbf{X}_{a,<i})\right), \qquad (2)$$

where $\theta$ are the trainable parameters, $\mathbf{X}_{q,<i}$ and $\mathbf{X}_{a,<i}$ are the query and, respectively, answer tokens located before the current predicted token $x_i$. $\mathbf{H}_v$ is obtained from Eq. 1. Note that the weights of LLM are shared between the two forward passes. The flow of gradients through $\mathbf{H}_v^c$ results in a single model that can both compress and generate answers by looking solely at the compressed tokens.

Intuitively, our algorithm can also be interpreted as a form of implicit chain-of-thought in the latent space [12], with the LLM "rephrasing" the content of the vision sequence in a condensed manner for itself. Notably, while the input and output spaces of the LLM are not perfectly aligned, they are sufficiently close to resulting in good alignment of the compressed representations in just

a few hundred iterations, making the whole training process efficient. That is, the compressed representations simultaneously lie in the input and output space of the LVLM.

## 3.3 Discriminative adaptation

Because the compressed representations in C&C lie simultaneously in both the input and output space of the LVLM (unlike previous approaches), this enables us to directly leverage them for CLIP-like discrimination in a zero-shot manner, as detailed in Sec. 3.1. However, in this case, the discriminant performance is suboptimal as there is no explicit loss to encourage the separability of concepts. To address this, and create a unified compressed representation suitable for both generative and discriminative tasks, we also propose to apply a contrastive loss over $\mathbf{H}_v^c$, at the output of the first forward pass. Importantly, this loss also turns out to enhance the generative ability of the model thanks to learning a better underlying representation.

Given a dataset consisting of paired image-text samples, the contrastive loss, for a given batch containing $B$ elements, is defined as:

$$\mathcal{L}_{\text{disc}} = \frac{1}{B} \sum_{k=1}^{b} (-\log \frac{\exp(s_v^{k,k})}{\sum_j \exp(s_v^{k,j})} - \log \frac{\exp(s_t^{k,k})}{\sum_j \exp(s_t^{j,k})}), \tag{3}$$

where $s_v^{k,j} = \cos\_\text{sim}(\mathbf{e}_v^k, \mathbf{e}_t^j)$ computes the cosine similarity between the $k$-th image and the $j$-th caption. $\mathbf{e}_v = \frac{1}{k'} \sum \mathbf{H}_v^c$ and, $\mathbf{e}_t = \frac{1}{k'} \sum \mathbf{H}_t^c$, respectively. $\mathbf{H}_t^c$ is computed analogously to $\mathbf{H}_v^c$ (Eq. 1), except that it encodes textual data instead of visual, *i.e.* $\mathbf{H}_t^c = f(\mathbf{H}_{query}, \mathbf{H}_p, \mathbf{H}_r)$. $\mathbf{H}_t^c$ is only used as part of the discriminative loss.

## 3.4 Overall training loss and data

The final model is trained using both losses, autoregressive and discriminative/contrastive:

$$\mathcal{L}_{\text{Total}} = \mathcal{L}_{\text{AR}} + \mathcal{L}_{\text{disc}}. \tag{4}$$

At a given iteration, depending on the sampled training data, the applicable losses are used. That is, for conversational data sampled from the LLaVA-665k dataset, we apply $\mathcal{L}_{\text{AR}}$. For data sampled from CC3M, we apply $\mathcal{L}_{\text{disc}}$. If a conversational sample also has a caption associated with it, both losses are applied within the same iteration. For efficiency, the sampler will group together such cases. This also ensures that we have sufficiently large batches for contrastive training. The sampler aims for a 1:1 ratio between discriminative and autoregressive.

## 3.5 Stage-specific adaptation

To enable efficient adaptation, we train our models using LoRA [14] adapters which restrict the weight updates to a low-rank representation, $\Delta W = BA, \Delta W \in \mathbb{R}^{d \times m}, B \in \mathbb{R}^{d \times r}$ and $A \in \mathbb{R}^{r \times m}$, with $r << \min(d, m)$. Although this works well, we use stage-specific adapters to further enhance the plasticity of the LVLM. We distinguish two stages that correspond to the two forward passes used during training: compression, which summarizes $\mathbf{H}_v$ into $\mathbf{H}_v^c$, and generation, which produces $\mathbf{X}_a$ given $\mathbf{H}_v^c$ and $\mathbf{X}_q$. Depending on the stage, different LoRA adapter weights $A$ and $B$ are used.

# 4 Results

## 4.1 Generative and Discriminative results with LLaVA-1.5

**Implementation details:** Our model is LLaVA-1.5 [34], consisting of a ViT-L@336px vision encoder [44] and Vicuna LLM [6] decoder/compressor. Unless otherwise stated, the models are trained for 10,000 iterations, using a batch size of 1024, AdamW [38] with no weight decay and a learning rate of $2e-4$ decayed to 0 using a cosine scheduler. All other layers remain frozen except for the LoRA adapters (rank = 64, $\alpha$ = 128). At a given iteration, depending on the loss (*i.e.* autoregressive or discriminative), we sample a batch either from LLaVA-665K [34] (for generative) or from CC3M [48] (for discriminative). The sampling ratio between the two is 1:1. The training runs were performed on 24 AMD MI300X GPUs using pytorch [43] and deepspeed [45].

**Generative benchmarks:** Following [34], we evaluate our approach on a diverse collection of datasets, mainly: GQA [18], MMB [37], MME [32], POPE [31], SQA [39], TextVQA [50],

Table 1: Comparison with various token reduction methods on vision-language understanding.

| Method | # Tokens | GQA | MMB | MME | POPE | SQA | TextVQA | VisWiz | VQAv2 |
|---|---|---|---|---|---|---|---|---|---|
| LLAVA-1.5 [34] | 576 | 62.0 | 64.3 | 1510.7 | 85.9 | 66.8 | 58.2 | 50.0 | 78.5 |
| PruMerge [47] | ≈32 | 57.2 | 60.9 | 1350.3 | 76.3 | 68.5 | **56.0** | 45.2 | 72.0 |
| TokenPacker [29] | 36 | 59.6 | 62.8 | 1440.9 | 83.3 | **71.0** | 53.2 | 50.2 | 75.0 |
| Matryoshka Multi. [3] | 36 | 60.3 | 64.8 | - | 85.5 | - | - | 52.8 | - |
| Matryoshka Query [16] | 36 | 58.8 | 63.4 | 1416.3 | 81.9 | 66.8 | - | 51.0 | 73.7 |
| QueCC [27] | 36 | 60.5 | 62.5 | 1442.0 | 84.5 | 70.6 | 53.3 | 50.1 | 75.8 |
| **C&C (Ours)** | 32 | **61.6** | 64.6 | **1472.1** | 85.9 | 68.5 | 55.8 | **53.1** | **77.1** |
| TokenPacker [29] | 16 | 58.9 | 62.7 | 1378.8 | 83.7 | 68.1 | 52.5 | 50.5 | 74.4 |
| Matryoshka Query [16] | 16 | 57.6 | 61.9 | 1408.5 | 80.8 | 67.5 | - | 49.8 | 71.1 |
| QueCC [27] | 16 | 59.0 | 62.2 | 1408.0 | 83.4 | **70.7** | 51.3 | 47.7 | 74.5 |
| **C&C (Ours)** | 16 | **61.0** | 64.4 | **1470.0** | 85.6 | 67.7 | **54.2** | 49.8 | **76.5** |
| TokenPacker [29] | 4 | 56.2 | 61.5 | 1347.6 | 81.7 | 68.5 | 49.2 | 45.7 | 70.5 |
| Matryoshka Query [16] | 4 | 53.0 | 56.5 | 1176.1 | 77.6 | 65.1 | - | 49.4 | 64.1 |
| QueCC [27] | 4 | 56.5 | 62.1 | 1390.3 | 81.8 | **68.6** | 48.7 | 45.0 | 70.6 |
| **C&C (Ours)** | 4 | **58.6** | **63.3** | **1403.0** | **84.3** | 67.7 | **52.5** | **51.6** | **74.5** |

Table 2: Zero-shot text-image retrieval accuracy on Flickr30K, COCO and nocaps.

| Method | **image** retrieval | | | | | | **text** retrieval | | | | | |
|---|---|---|---|---|---|---|---|---|---|---|---|---|
| | Flickr30K | | COCO | | nocaps | | Flickr30K | | COCO | | nocaps | |
| | R@1 | R@10 | R@1 | R@10 | R@1 | R@10 | R@1 | R@10 | R@1 | R@10 | R@1 | R@10 |
| Contrastive approaches | | | | | | | | | | | | |
| CLIP (ViT-L) [44] | 67.3 | 93.3 | 37.0 | 71.5 | 48.6 | 85.7 | 87.2 | 99.4 | 58.1 | 87.8 | 70.0 | 96.2 |
| BLIP (ViT-L) [25] | 70.0 | 95.2 | 48.4 | 83.2 | 62.3 | 93.4 | 75.5 | 97.7 | 63.5 | 92.5 | 72.1 | 97.7 |
| BLIP2 (ViT-L) [26] | 74.5 | 97.2 | 50.0 | 86.1 | 63.0 | 93.8 | 86.1 | 99.4 | 63.0 | 93.1 | 74.4 | 98.3 |
| OpenCLIP (ViT-G/14) [46] | 77.8 | 96.9 | 48.8 | 81.5 | 63.7 | 93.2 | 91.5 | 99.6 | 66.3 | 91.8 | 81.0 | 98.7 |
| OpenCLIP (ViT-BigG/14) [46] | 79.5 | 97.1 | 51.3 | 83.0 | 65.1 | 93.5 | 92.9 | 97.1 | 67.3 | 92.6 | 82.3 | 98.8 |
| EVA-02-CLIP (ViT-E/14+) [51] | 78.8 | 96.8 | 51.1 | 82.7 | 64.5 | 92.9 | 93.9 | 99.8 | 68.8 | 92.8 | 83.0 | 98.9 |
| EVA-CLIP [52] | 80.3 | 97.2 | 52.0 | 82.9 | 65.3 | 93.2 | 94.5 | 99.7 | 70.1 | 93.1 | 83.5 | 98.6 |
| LVLM-based approaches | | | | | | | | | | | | |
| LLaVA-1.5-7B [34] | 59.6 | 89.3 | 34.4 | 69.6 | 46.9 | 83.3 | 65.6 | 92.3 | 35.6 | 70.5 | 52.1 | 88.1 |
| E5-V (LLaVA-1.5-7B) [20] | 76.7 | 96.9 | 48.2 | 82.1 | 62.0 | 93.0 | 86.6 | 99.0 | 57.4 | 88.4 | 71.9 | 97.0 |
| VLM2Vec (Mistral-7B) [21] | 80.1 | 97.3 | 52.0 | 85.6 | 65.9 | 94.5 | 90.3 | 99.6 | 68.2 | 93.2 | 79.2 | 98.5 |
| **C&C (Ours)** (LLaVA-1.5-7B) | **83.8** | **98.5** | **56.8** | **86.6** | **70.2** | **96.1** | 94.3 | **99.9** | 72.9 | 94.4 | 85.7 | 99.5 |

VisWiz [11] and VQAv2 [10]. To ensure fairness, in all cases, we fully align the test-time settings and processing with [34]. In addition to this, we also evaluate our approach for captioning on MS-COCO [33], Flickr30k [58] and NoCaps [1], comparing it to token reduction methods that have models openly available. See supplementary material for results on TextCaps [49].

When comparing our approach with the state-of-the-art token reduction methods for visual-language understanding, as the results from Tab. 1 show, we set a new best result, outperforming prior works using 2.25× fewer tokens (16 vs 36). Our results for 32 and even 16 tokens nearly match the uncompressed LlaVA [34] baseline.

Similarly, when evaluated for zero-shot captioning (Tab. 3), our approach matches LLaVA's accuracy, significantly outperforming prior methods. This suggests that the proposed approach encodes more information in its compressed tokens. We note that LLaVA saw some MS-COCO images during training; hence, the MS-COCO evaluation is not fully zero-shot for all methods listed.

**Discriminative benchmarks:** We evaluate our model on a diverse set of retrieval benchmarks: Flicr30k [58], MS-COCO [33], NoCaps [1] and SugarCrepe [13], against state-of-the-art two-tower independent models. The last one measures the compositional capabilities of the model, an area where CLIP and CLIP-like models tend to underperform.

As Tab. 2 shows, we match and outperform several state-of-the-art contrastive models including larger models, i.e. EVA-CLIP (8B vs. 7.06B), despite using 3 orders of magnitude fewer samples for training (2,700M for EVA-CLIP vs. ∼3M for ours). A similar trend can be observed when evaluated for compositionality on SugarCreppe (Tab. 4). Interestingly, models derived from LVLMs (e.g., E5-V and ours) demonstrate superior compositionality. This suggests that the LVLM in discriminative mode inherits the strong vision-language understanding of the underlying generative model.

Table 3: Comparison with various token reduction/compression methods on image captioning in terms of CIDEr score. See supplimentary material for results using additional metrics.

| Method | # Tokens | Flickr30K | COCO | nocaps |
|---|---|---|---|---|
| LLAVA-1.5 [34] | 576 | 81.2 | 115.4 | 105.3 |
| PruMerge [47] | ≈32 | 36.3 | 66.3 | 58.6 |
| Matryoshka Multi. [3] | 36 | 68.7 | 102.2 | 93.6 |
| Matryoshka Query [16] | 36 | 69.5 | 101.3 | 90.0 |
| C&C (Ours) | 32 | **78.9** | **113.1** | **105.9** |
| Matryoshka Query [16] | 16 | 65.2 | 99.2 | 90.0 |
| C&C (Ours) | 16 | **78.2** | **112.0** | **104.7** |
| Matryoshka Query [16] | 4 | 47.5 | 81.0 | 63.2 |
| C&C (Ours) | 4 | **74.5** | **111.4** | **103.4** |

Table 4: Comparison with state-of-the-art on the SugarCrepe compositionality benchmark.

| Method | Params (B) | Replace | Swap | Add |
|---|---|---|---|---|
| Contrastive approaches | | | | |
| NegCLIP [61] | 0.15 | 85.0 | 75.3 | 85.8 |
| CLIP (ViT-L) [44] | 0.43 | 79.5 | 61.3 | 74.9 |
| BLIP (ViT-L) [25] | 0.23 | 82.4 | 71.7 | 88.6 |
| BLIP2 (ViT-L) [26] | 1.17 | 85.7 | 63.8 | 89.9 |
| OpenCLIP (ViT-G/14) [46] | 1.37 | 84.4 | 67.1 | 86.8 |
| OpenCLIP (ViT-BigG/14) [46] | 2.54 | 86.5 | 68.9 | 88.4 |
| EVA-02-CLIP (ViT-E/14+) [51] | 5.04 | 86.6 | 70.7 | 87.9 |
| EVA-CLIP [52] | 8.22 | 85.9 | 70.4 | 86.7 |
| LVLM-based approaches | | | | |
| LLaVA-1.5-7B [34] | 7.06 | 81.9 | 59.9 | 64.7 |
| E5-V (LLaVA-1.5-7B) [20] | 7.06 | 88.0 | 63.5 | 90.8 |
| VLM2Vec (Mistral-7B) [21] | 7.3 | 89.3 | 67.7 | 91.7 |
| **C&C (Ours)** (LLaVA-1.5-7B) | 7.06 | **90.1** | **77.9** | **94.2** |

## 4.2 Visual RAG results with LLaVA-OneVision

Our method offers distinct advantages for Vision-based RAG, stemming from its upfront indexing step and a unified representation for both retrieval and generation. This allows us to use a single model, in contrast to prior methods, which require separate models for retrieval and generation. Below, we compare our approach with the state-of-the-art following the protocol of VisRAG [60].

**Implementation details:** Since LLaVA-1.5 model is not suitable for high-resolution image analysis, we adopted the improved LLaVA-OneVision-0.5B [23]. We keep the previous hyperparameters fixed, changing only the training data to reflect the nature of the task and model. In particular, for the generative objective, we use the same collection of vision-language datasets encompassing 3.2M samples introduced in [23] and previously used to train the LLaVA-OneVision model that we start from. For the discriminative objective, to allow for fair comparisons, we use the same data as in [60].

**Comparison with state-of-the-art:** We report results on all datasets from [60] (*i.e.* ArxivQA [28], ChartQA [40], DocVQA [54], InfoVQA [41], PlotQA [42], SlideVQA [53]), first in terms of retrieval and then in terms of retrieval-augmented generation (RAG). For retrieval, as Tab. 11 shows, despite using a $3.8\times$ smaller model, capable of performing generation too, we match and outperform the state-of-the-art VisRAG-Ret [60]. For RAG evaluation, for the generator we consider the following options: MiniCPM-V 2.6 [15], LLaVA-OV-0.5B (original), and our C&C (based on LLaVA-OV-0.5B). For the retriever: VisRAG-Ret [60] and C&C. Notice that in our case, we use the same model for both generation and retrieval. We report in Tab. 6 all combinations formed by them. Our approach gets close to the original LLaVA-OV-0.5B model, using $24\times$ fewer vision tokens. Furthermore, for larger generators (MiniCPM-V 2.6), our improved retriever translates into better RAG performance.

Table 5: Overall retrieval performance in MRR@10. Corresponding Recall@10 performance can be found in the supplementary material. All prior methods results are taken from [60].

| Model | # Param. | ArxivQA | ChartQA | DocVQA | InfoVQA | PlotQA | SlideVQA | Average |
|---|---|---|---|---|---|---|---|---|
| MiniCPM (OCR) [15] | 2.72B | 58.43 | 77.74 | 72.54 | 83.45 | 64.78 | 91.74 | 74.78 |
| MiniCPM (Captioner) [15] | 2.72B | 56.15 | 74.06 | 67.57 | 81.22 | 55.43 | 84.27 | 69.78 |
| SigLIP [2023] | 0.88B | 59.16 | 81.34 | 64.60 | 74.59 | 61.32 | 89.08 | 71.68 |
| ColPali [2024] | 2.92B | 72.50 | 73.49 | **82.79** | 81.15 | 55.32 | **93.99** | 76.54 |
| VisRAG-Ret [60] | 3.43B | **75.11** | 76.63 | 75.37 | 86.37 | 62.14 | 91.85 | 77.91 |
| **C&C (Ours)** | 0.89B | 74.63 | **87.04** | 74.79 | **86.40** | **68.73** | 90.99 | **80.38** |

Table 6: Overall generation performance in accuracy (%) using two retrievers: VisRAG-Ret and C&C (Ours). Our variant uses a LLaVA-OV-0.5B model and compresses each patch from 768 to 32 tokens. Note that unlike prior works, our model can perform both retrieval and generation.

| Generator | Retriever | ArxivQA | ChartQA | DocVQA | InfoVQA | PlotQA | SlideVQA | Average |
|---|---|---|---|---|---|---|---|---|
| MiniCPM-V 2.6 | VisRAG-Ret | **66.67** | 46.88 | **54.31** | 63.34 | 47.57 | 50.54 | 54.89 |
| MiniCPM-V 2.6 | **C&C (Ours)** | 66.42 | **54.69** | 53.33 | **64.19** | **49.19** | **50.71** | **56.42** |
| LLaVA-OV-0.5B | VisRAG-Ret | 46.81 | 31.25 | 24.72 | 38.01 | 21.06 | 33.21 | 32.51 |
| C&C (Ours) | VisRAG-Ret | 46.32 | 34.38 | 22.92 | 30.91 | 23.26 | 28.04 | 30.97 |
| LLaVA-OV-0.5B | **C&C (Ours)** | 45.83 | 34.38 | 25.00 | 38.18 | 21.18 | 32.68 | 32.88 |
| **C&C (Ours)** | **C&C (Ours)** | 45.93 | 39.06 | 22.50 | 30.24 | 22.92 | 28.39 | 31.49 |

## 5 Ablation studies & analysis

**Impact of each loss function:** As detailed in Secs. 3.2 and 3.3, our models are trained using two losses: one autoregressive, applied after the second forward pass, and one contrastive, applied

Table 7: Effect of generative and discriminative losses for generation (MMB, MME, TextVQA) and retrieval (Flickr30K, MS-COCO).

| Method | MMB | MME | Text VQA | Flickr30K T2I | Flickr30K I2T | MS-COCO T2I | MS-COCO I2T |
|---|---|---|---|---|---|---|---|
| Discrim. | 46.2 | 624.3 | 13.5 | 84.3 | 94.8 | 56.3 | 73.2 |
| Generative | 64.1 | 1420.1 | 54.2 | 61.3 | 76.0 | 33.9 | 47.0 |
| Both | 64.4 | 1470.0 | 54.2 | 83.8 | 94.4 | 56.8 | 70.2 |

Table 8: Single vs. stage-specific LoRA v.s full finetuning for generation (MMB, MME, TextVQA) and retrieval (Flickr30K, MS-COCO).

| Method | MMB | MME | Text VQA | Flickr30K T2I | Flickr30K I2T | MS-COCO T2I | MS-COCO I2T |
|---|---|---|---|---|---|---|---|
| Fine-tuning | 64.3 | 1413.1 | 52.9 | 83.1 | 94.0 | 56.2 | 70.4 |
| Single LoRA | 64.3 | 1410.5 | 51.8 | 83.8 | 94.1 | 56.5 | 69.9 |
| Stage LoRA | 64.4 | 1470.0 | 54.2 | 83.8 | 94.4 | 56.8 | 70.2 |

over the compressed tokens, after the first pass. In Tab. 7, we report results for the LLaVA model evaluated using 16 tokens on generative and discriminative tasks. Intuitively, training solely with the discriminative loss (1st row) results in degraded generative performance, as no alignment between the input and output space of the LLM is performed. Moreover, discriminative losses applied over short captions tend to focus on coarse details, missing out on finer-grained details. Conversely, applying only the generative loss (2nd row) results in degraded retrieval abilities, as no loss explicitly encourages concept separation. We note that the longer the training scheduler is, the more pronounced these degradations are for the two cases.

Finally, combining the two losses (3rd row) results in the best performance across the board. Notice that the two losses are complementary when applied jointly and boost the model's accuracy on both sets of benchmarks.

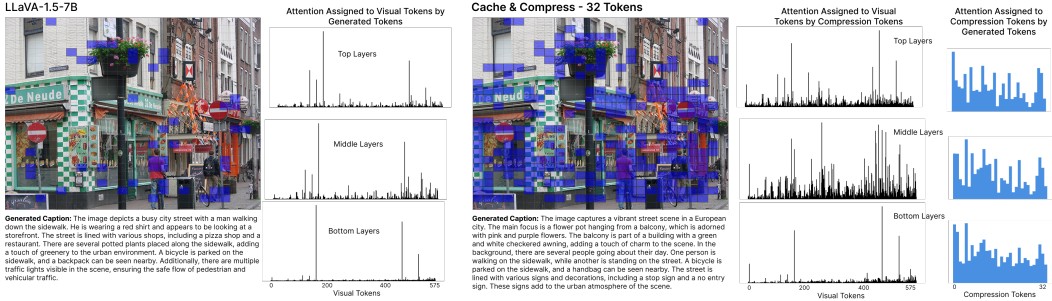

Figure 4: Visualization of attention weights assigned to the 576 visual tokens and the 32 compressed tokens. On the left, we show the cumulative weights assigned to each visual token by the generated tokens for the base model. For C&C, on the right, we first display the per-visual-token weights assigned by the summary tokens during the 1st forward pass for compression. We then show the weights assigned to the compressed tokens by the generated ones during the 2nd forward pass.

**Single vs. stage-specific LoRA vs. full fine-tuning:** Herein, we compare the effect of training using (a) a single shared LoRA adapter, (b) stage-specific adapters, as proposed in Sec. 3.5, and (c) full fine-tuning. We present the results of these choices in Tab. 8. The best results are obtained using the stage-specific adapters. The fine-tuning run suffers from overfitting to some extent, and its larger training cost makes optimization more difficult.

Fig. 3 further solidifies the need for stage-specific LoRAs, as the optimal representations required during compression (first forward pass) vs downstream inference (second forward pass) are different, especially for earlier layers.

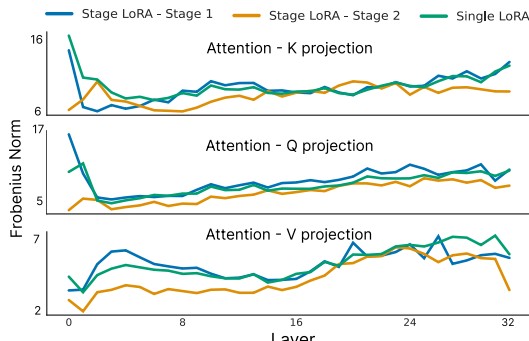

Figure 3: The norm of the learned LoRA weights adjustment $\Delta W = BA$ for a model trained with either a single LoRA or stage-specific LoRAs.

**Double forward vs single forward:** To showcase the importance of our "double-forward pass" training strategy, we conducted the following experiment: instead of using the LLM itself to compress the vision summary tokens, we use the CLIP vision encoder only. In this case, the loss is directly applied after the LLM, as in LLaVA, using a single forward pass. As shown in Tab. 9, this baseline (1st row) vs ours (2nd row) performs significantly worse.

**How does the model's behavior change?** To shed light on the changes the model undergoes to act as a self-compressor, we analyze the attention patterns before and after our fine-tuning. The results of this visual analysis are presented in Fig. 4. Looking on the left side, we can observe that LLaVA exhibits a sparse attention pattern across all layers, particularly early on. In contrast, during self-compression, our model attends to all visually important parts of the image, having a significantly denser attention pattern at all layers. Intuitively, in the first case, as the model has access to all tokens, during generation, the model can peek back at the vision tokens as needed. In contrast, during compression, the LLM must ensure that all visually important details are stored in the compressed representation. Finally, on the right-most part of the figure, we showcase the attention pattern between the generated tokens and the compressed representation obtained during text generation from compressed representations. We observe that early and late summary tokens generally receive higher attention weights.

**Efficiency analysis:** Unlike prior works that perform the compression on-the-fly, our approach offloads the compression cost to a dedicated up-front indexing stage. This disentaglement allows for a more expensive and highly accurate compressor that is run ahead of time. This scenario is aligned

Table 9: Double vs single forward pass for generation (MMB, MME, TextVQA) and retrieval (Flickr30K, MS-COCO).

| Method | MMB | MME | Text VQA | Flickr30K T2I | Flickr30K I2T | MS-COCO T2I | MS-COCO I2T |
|---|---|---|---|---|---|---|---|
| Single-fwd stage | 61.3 | 1305.0 | 42.1 | 80.8 | 92.1 | 53.1 | 66.2 |
| Double-fwd (Ours) | 64.4 | 1470.0 | 54.2 | 83.8 | 94.4 | 56.8 | 70.2 |

with RAG and on-device deployment, where most images from a gallery can be indexed overnight. While we note that some of the methods we compare with could be run offline too (*i.e.* [16, 3]), they (a) don't make this distinction, (b) have significantly worse accuracy, and (c) produce representations unsuitable for retrieval.

With this in mind, in Fig. 5 we report the FLOPs count for a series of state-of-the-art methods during the indexing (caching) and generation phase. The FLOPs count is estimated as in [22, 27] under the following setting: only the prefilling FLOPs are captured, all token compression methods use 16 tokens, the LLaVA-1.5 baseline uses all 576 tokens, for QueCC, which performs query-dependent compression, we assume an average query length of 25 tokens. As the figure shows, our method is the only one to

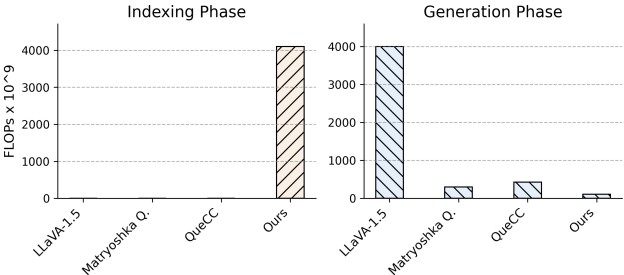

Figure 5: FLOPs estimate for various methods for the indexing and generation phase.

leverage a slower but highly accurate compressor during an indexing phase with a compute cost similar to running the baseline model. During generation, our approach is the fastest as it directly loads the cached tokens, bypassing the need to recompute the vision tokens using the vision encoder and a compression module. In contrast, the current state-of-the-art approach, QueCC [27], requires nearly $2\times$ more FLOPs due to the dependency on the user query/instruction for compression. Moreover, from a storage point of view, in [27], all V (576) tokens must be stored or, alternatively, recomputed if an image from the database is queried again.

**Limitations and broader impact:** As our work reduces the inference cost, it allows the deployment of highly performant LVLMs and RAG systems on-device, reducing costs and democratizing the use of AI. In terms of limitations, our work builds on top of existing pre-trained LVLMs. As our goal is to explicitly preserve their characteristics, any potential biases present in the original data and model are likely to propagate to ours too. Therefore, we recommend caution before deploying such models. Moreover, while the proposed method is well-suited for on-device deployment and RAG systems, it's less so for scenarios that don't allow for offline preprocessing and caching.

## 6 Conclusions

In this work, we introduced C&C, a novel LVLM visual token compression approach that uses the LVLM itself to compress the visual information in a task-agnostic manner, which is trained using a

new "double-forward pass" training strategy. This results in a compressed visual representation that is simultaneously suitable for (a) generative and (b) discriminative tasks, (c) is nearly lossless, and (d) is storage-efficient. Performance-wise, for generative tasks, we offer a $2\times$ higher compression rate without compromising the generative capabilities, setting a new state-of-the-art. For discriminative tasks, we also set a new state-of-the-art result on image retrieval and compositionality.

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

# A  Additional results and comparisons

**Results for larger LLaVA-1.5 models:**   In the main manuscript, we conduct experiments using a LLaVA-1.5 (7B) model. Herein, we validate how our approach behaves when using a larger model, *i.e.* a LLaVA-1.5 (13B). As the results from Table 10 show, the proposed method nearly matches the full LLaVA model's accuracy using only 16 and even 4 tokens in this case, too.

Table 10: Token compression performance on vision-language understanding tasks using a LLaVA-1.5 13B model.

| Method | # Tokens | GQA | MMB | MME | POPE | SQA | TextVQA | VisWiz | VQAv2 |
|---|---|---|---|---|---|---|---|---|---|
| LLAVA-1.5 [34] | 576 | 63.3 | 67.7 | 1531.0 | 86.2 | 71.6 | 61.3 | 53.6 | 80.0 |
| C&C (Ours) | 32 | 62.2 | 67.6 | 1465.1 | 85.3 | 72.5 | 59.6 | 54.0 | 78.7 |
| C&C (Ours) | 16 | 61.8 | 67.3 | 1473.5 | 85.0 | 72.4 | 57.5 | 54.2 | 78.4 |
| C&C (Ours) | 4 | 59.9 | 66.4 | 1390.1 | 84.4 | 71.1 | 53.6 | 52.7 | 75.9 |

**Visual RAG results with Larger LLaVA-OneVision models**   In addition to the Visual RAG results from the main manuscript, which use a 0.5B LLaVA-OV skew, herein we report results for a larger 7B model (*i.e.* LLaVA-OV-7B). As shown in Table 11 and 12, for retrieval, our approach outperforms prior works in terms of both MRR@10 and Recall@10, showing good scaling with respect to the model size and establishing a new state-of-the-art result.

Table 11: Overall retrieval performance in terms of MRR@10. All prior methods results are taken from [60].

| Model | # Param. | ArxivQA | ChartQA | DocVQA | InfoVQA | PlotQA | SlideVQA | Average |
|---|---|---|---|---|---|---|---|---|
| MiniCPM (OCR) [15] | 2.72B | 58.43 | 77.74 | 72.54 | 83.45 | 64.78 | 91.74 | 74.78 |
| MiniCPM (Captioner) [15] | 2.72B | 56.15 | 74.06 | 67.57 | 81.22 | 55.43 | 84.27 | 69.78 |
| SigLIP [2023] | 0.88B | 59.16 | 81.34 | 64.60 | 74.59 | 61.32 | 89.08 | 71.68 |
| ColPali [2024] | 2.92B | 72.50 | 73.49 | 82.79 | 81.15 | 55.32 | 93.99 | 76.54 |
| VisRAG-Ret [60] | 3.43B | 75.11 | 76.63 | 75.37 | 86.37 | 62.14 | 91.85 | 77.91 |
| **C&C** (LLaVA-OV-0.5B) **(Ours)** | 0.89B | 74.63 | 87.04 | 74.79 | 86.40 | 68.73 | 90.99 | 80.38 |
| **C&C** (LLaVA-OV-7B) **(Ours)** | 7.0B | **83.65** | **90.56** | **85.77** | **91.97** | **71.41** | **95.07** | **86.41** |

Table 12: Overall retrieval performance in terms of Recall@10. All prior methods results are taken from [60].

| Model | # Param. | ArxivQA | ChartQA | DocVQA | InfoVQA | PlotQA | SlideVQA | Average |
|---|---|---|---|---|---|---|---|---|
| MiniCPM (OCR) [15] | 2.72B | 69.36 | 88.89 | 87.14 | 94.15 | 90.61 | 96.85 | 87.83 |
| MiniCPM (Captioner) [15] | 2.72B | 69.00 | 85.71 | 84.26 | 94.29 | 84.24 | 93.08 | 85.10 |
| SigLIP [2023] | 0.88B | 73.90 | 92.06 | 83.08 | 93.04 | 89.57 | 94.15 | 87.63 |
| ColPali [2024] | 2.92B | 82.72 | 88.89 | 94.75 | 94.43 | 80.30 | 97.21 | 89.72 |
| VisRAG-Ret [60] | 3.43B | 87.25 | 90.48 | 91.20 | 97.08 | 89.80 | 97.39 | 92.20 |
| **C&C** (LLaVA-OV-0.5B) **(Ours)** | 0.89B | 84.64 | 92.06 | 91.71 | 97.21 | 93.51 | 96.67 | 92.97 |
| **C&C** (LLaVA-OV-7B) **(Ours)** | 7.0B | **93.38** | **98.41** | **96.45** | **98.75** | **94.67** | **98.47** | **96.69** |

**Additional discriminative comparisons with other token-summarization approaches.** We note that our approach is the only one that compresses the vision tokens into a representation suitable for both generative and discriminative tasks, requiring no additional forward passes. However, herein, for completeness, we evaluate on our suite of discriminative tasks the current state-of-the-art token compression models that offered pretrained models. This is achieved by following the zero-shot setup described in the main manuscript in Section 3.1 and [20]. Unsurprisingly, as the results from Table 13 and 14 show, our approach significantly surpasses the other methods we compare with.

Table 13: Zero-shot text-image retrieval accuracy on Flickr30K, COCO and nocaps. Only our approach is specialised for both retrieval and generation, hence, except for our method, all other results are in a zero-shot manner following the protocol described in the main manuscript in Section 3.1 and [20].

| Method | Tokens | **image** retrieval | | | | | | **text** retrieval | | | | | |
|---|---|---|---|---|---|---|---|---|---|---|---|---|---|
| | | Flickr30K | | COCO | | nocaps | | Flickr30K | | COCO | | nocaps | |
| | | R@1 | R@10 | R@1 | R@10 | R@1 | R@10 | R@1 | R@10 | R@1 | R@10 | R@1 | R@10 |
| LLaVA-1.5-7B [34] | 576 | 59.6 | 89.3 | 34.4 | 69.6 | 46.9 | 83.3 | 65.6 | 92.3 | 35.6 | 70.5 | 52.1 | 88.1 |
| PruMerge [47] | 18 | 34.7 | 67.9 | 18.4 | 47.9 | 25.8 | 62.7 | 38.3 | 74.3 | 19.8 | 49.9 | 28.2 | 65.2 |
| Matryoshka Multi. [3] | 16 | 57.9 | 88.5 | 34.1 | 69.7 | 45.5 | 83.2 | 63.8 | 91.7 | 36.4 | 72.5 | 48.0 | 86.2 |
| Matryoshka Query [16] | 16 | 53.6 | 85.9 | 29.8 | 65.4 | 40.5 | 80.0 | 59.4 | 90.3 | 34.1 | 69.6 | 45.4 | 84.7 |
| C&C (Ours) | 16 | **83.8** | **98.5** | **59.0** | **88.6** | **72.3** | **96.5** | **94.3** | **99.9** | **72.9** | **94.4** | **85.7** | **99.5** |

Table 14: Comparison on the SugarCrepe compositionality benchmark.

| Method | Tokens | Replace | | | Swap | | Add | |
|---|---|---|---|---|---|---|---|---|
| | | Object | Attribute | Relation | Object | Attribute | Object | Attribute |
| LLaVA-1.5-7B [34] | 576 | 88.0 | 81.6 | 76.1 | 60.9 | 58.8 | 67.0 | 62.4 |
| PruMerge [47] | 18 | 88.0 | 74.4 | 69.7 | 62.5 | 57.3 | 81.4 | 66.0 |
| Matryoshka Multi. [3] | 16 | 90.3 | 81.4 | 80.1 | 70.2 | 67.9 | 75.7 | 75.8 |
| Matryoshka Query [16] | 16 | 89.3 | 81.4 | 79.2 | 70.6 | 64.7 | 73.8 | 73.6 |
| C&C (Ours) (LLaVA-1.5-7B) | 7.06 | **98.1** | **89.5** | **82.7** | **77.8** | **78.1** | **95.3** | **93.1** |

**Additional zero-shot image captioning evaluations:** In addition to the evaluation from the main manuscript, herein, we evaluate our approach for zero-shot captioning on TextCaps [49], a dataset for image captioning with reading comprehension. As the results from Table 15 show, we generally match the full-tokens LLaVA's model performance. Importantly, our results remain stable as the number of compressed tokens decreases.

**In-depth evalution results for captioning:** In the main manuscript we report results for image captioning solely in terms of CIDEr score. For completeness, in Table 16 we also report Bleu@4 (B@4), METEOR (MET.), and ROUGE. The conclusions hold across all metrics.

**In-depth evalution results for captioning:** In the main manuscript we report results for image captioning solely in terms of CIDEr score. For completeness, in Table 16 we also report Bleu@4 (B@4), METEOR (MET.), and ROUGE. The conclusions hold across all metrics.

# B  Additional ablation studies and analyses

**What do the compressed tokens encode?** The compressed representation gradually encodes, from left to right, coarser to finer-grained concepts. This effect can be observed in Fig. 6, where, as the

Table 15: Comparison with various token compression methods on TextCaps dataset for image captioning in terms of BLEU-4 (B@4), CIDEr score, METEOR (MET.) and ROUGE-L.

| Method | Tokens | B@4 | CIDEr | MET. | ROUGE |
|---|---|---|---|---|---|
| LLAVA-1.5 [34] | 576 | 27.1 | 90.4 | 21.9 | 46.2 |
| PruMerge [47] | ≈32 | 17.6 | 62.8 | 17.0 | 39.7 |
| Matryoshka Multi. [3] | 36 | 25.1 | 94.8 | 23.0 | 46.3 |
| Matryoshka Query [16] | 36 | 21.0 | 70.0 | 19.9 | 42.6 |
| C&C (Ours) | 32 | 26.5 | 90.6 | 22.4 | 46.1 |
| Matryoshka Query [16] | 16 | 20.1 | 62.5 | 19.3 | 41.7 |
| C&C (Ours) | 16 | 26.4 | 90.5 | 22.5 | 46.3 |
| Matryoshka Query [16] | 4 | 15.2 | 42.0 | 16.5 | 37.4 |
| C&C (Ours) | 4 | 25.4 | 86.1 | 22.0 | 45.7 |

Table 16: Comparison with various token reduction/compression methods on image captioning.

| Method | # Tokens | Flickr30K | | | | COCO | | | | nocaps | | | |
|---|---|---|---|---|---|---|---|---|---|---|---|---|---|
| | | B@4 | CIDEr | MET. | ROUGE | B@4 | CIDEr | MET. | ROUGE | B@4 | CIDEr | MET. | ROUGE |
| LLAVA-1.5 [34] | 576 | 30.6 | 81.2 | 25.0 | 53.4 | 32.9 | 115.4 | 27.7 | 56.3 | 42.9 | 105.3 | 28.9 | 59.8 |
| PruMerge [47] | ≈32 | 18.5 | 36.3 | 15.7 | 40.2 | 18.5 | 66.3 | 18.8 | 44.9 | 25.9 | 58.6 | 20.0 | 47.8 |
| Matryoshka Multi. [3] | 36 | 25.4 | 68.7 | 24.1 | 49.9 | 27.7 | 102.2 | 27.2 | 53.3 | 36.8 | 93.6 | 28.0 | 56.5 |
| Matryoshka Query [16] | 36 | 26.4 | 69.5 | 23.1 | 50.0 | 28.0 | 101.3 | 26.2 | 52.7 | 36.2 | 90.0 | 26.8 | 55.8 |
| C&C (Ours) | 32 | **30.0** | **78.9** | **25.2** | **52.9** | **31.5** | **113.1** | **27.9** | **55.6** | **42.5** | **105.9** | **29.2** | **59.6** |
| Matryoshka Query [16] | 16 | 24.8 | 65.2 | 22.7 | 49.0 | 27.6 | 99.2 | 26.0 | 52.5 | 36.2 | 90.0 | 26.8 | 55.8 |
| C&C (Ours) | 16 | **29.0** | **78.2** | **25.3** | **52.7** | **31.0** | **112.0** | **27.9** | **55.4** | **42.0** | **104.7** | **29.3** | **59.5** |
| Matryoshka Query [16] | 4 | 20.1 | 47.5 | 19.8 | 44.5 | 23.2 | 81.0 | 23.0 | 48.6 | 28.4 | 63.2 | 21.1 | 49.5 |
| C&C (Ours) | 4 | **28.4** | **74.5** | **24.8** | **51.8** | **31.1** | **111.4** | **27.9** | **55.4** | **41.1** | **103.4** | **29.0** | **59.1** |

Table 17: Comparison with state-of-the-art on the SugarCrepe compositionality benchmark.

| Method | Params (B) | Replace | | | Swap | | Add | |
|---|---|---|---|---|---|---|---|---|
| | | Object | Attribute | Relation | Object | Attribute | Object | Attribute |
| | | Contrastive approaches | | | | | | |
| NegCLIP [61] | 0.15 | 92.7 | 85.9 | 76.5 | 75.2 | 75.4 | 88.8 | 82.8 |
| CLIP (ViT-B) [44] | 0.15 | 90.9 | 80.1 | 69.2 | 61.4 | 64.0 | 77.2 | 68.8 |
| CLIP (ViT-L) [44] | 0.43 | 94.1 | 79.2 | 65.2 | 60.2 | 62.3 | 78.3 | 71.5 |
| BLIP (ViT-L) [25] | 0.23 | 96.5 | 81.7 | 69.1 | 66.6 | 76.8 | 92.0 | 85.1 |
| BLIP2 (ViT-L) [26] | 1.17 | 97.6 | 81.7 | 77.8 | 62.1 | 65.5 | 92.4 | 87.4 |
| OpenCLIP (ViT-G/14) [46] | 1.37 | 95.8 | 85.0 | 72.4 | 63.0 | 71.2 | 91.5 | 82.1 |
| OpenCLIP (ViT-BigG/14) [46] | 2.54 | 96.6 | 87.9 | 74.9 | 62.5 | 75.2 | 92.2 | 84.5 |
| EVA-02-CLIP (ViT-E/14+) [51] | 5.04 | 97.1 | 88.5 | 74.2 | 67.3 | 74.1 | 91.8 | 83.9 |
| EVA-CLIP [52] | 8.22 | 96.4 | 86.6 | 74.8 | 66.1 | 74.6 | 91.3 | 82.0 |
| | | LVLM-based approaches | | | | | | |
| LLaVA-1.5-7B [34] | 7.06 | 88.0 | 81.6 | 76.1 | 60.9 | 58.8 | 67.0 | 62.4 |
| E5-V (LLaVA-1.5-7B) [20] | 7.06 | 95.8 | 86.6 | 81.6 | 62.9 | 64.0 | 93.5 | 88.0 |
| VLM2Vec (Mistral-7B) [21] | 7.3 | 97.2 | 89.0 | 81.7 | 62.9 | 72.5 | 94.7 | 88.6 |
| C&C (Ours) (LLaVA-1.5-7B) | 7.06 | **98.1** | **89.5** | **82.7** | **77.8** | **78.1** | **95.3** | **93.1** |

number of tokens increases, the caption generated correctly captures more elements present in the photo, importantly reducing hallucinations. This effect is also corroborated in Fig. 7. There, we mask out different groups of (4 and 8) tokens, quantitatively measuring the impact of this: earlier tokens induce larger drops in performance (*e.g.* masking the first 8 tokens reduces performance by 10%). However, the performance does not drop to (near) 0, which suggests that there is also some degree of redundancy. The observed behavior can largely be attributed to the causal attention masking used by the LVLM, which encourages a directional information distribution.

**Full-attention vs causal:** Vicuna, and hence LLaVA, much like the rest of the generative LLMs, employs causal attention masking in order to restrict the past states from attending the future ones. While necessary for autoregressive modeling, it's unclear why it would be for vision token compression, too, as there is no preferential direction for image processing. Hence, herein, we explore the effect of changing the attention pattern from causal to bidirectional (*i.e.* full) attention for the compression forward pass, while keeping it causal for the subsequent answer generation ones. In this instance, the stage (*i.e.* compression vs generative) specific LoRAs also take the role of adjusting

**Prompt: Generate a detailed caption of the image.**

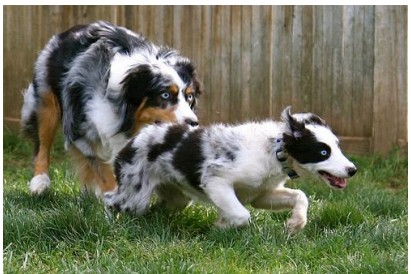

**Fwd2Bot**

**#Token = 32**

In the image, there are two dogs, one adult and one puppy, playing together in a grassy area. The adult dog is positioned on the left side of the image, while the puppy is on the right. The adult dog is larger and has a black and white coat, while the puppy is smaller and has a black and white coat with a blue collar. Both dogs are running towards each other, indicating a playful interaction between them. The background of the image is a wooden fence, providing a natural setting for the dogs' playtime.

**LLaVA-1.5-7B**

The image features a dog and a puppy running together in a grassy field. The dog is positioned on the left side of the image, while the puppy is on the right side. Both dogs appear to be enjoying their time outdoors. In the background, there is a fence visible, likely marking the boundary of the field. The scene captures the playful and energetic nature of the dogs as they run and interact with each other.

**#Token = 16**

In the image, there are two dogs in a grassy area. The larger dog, which is brown and white, is standing on the left side of the image. The smaller dog, which is black and white, is standing on the right side of the image. Both dogs are looking in the same direction, possibly at something off-camera. The larger dog appears to be walking towards the smaller dog, creating a sense of movement in the scene.

**#Token = 4**

In the image, there are two dogs, one black and white and the other brown and white, playing together in a grassy field. The black and white dog is positioned on the left side of the image, while the brown and white dog is on the right. They are both facing each other, engaged in a playful interaction. The field they are in is lush and green, providing a perfect environment for their playtime.

Figure 6: Captioning with variable number of summary tokens.

the attention pattern and information flow. Analyzing the results from Table 18 we can observe performance gains for discriminative tasks and degradation for generative ones. This suggests that a direct finetuning under a different attention pattern is suboptimal, likely requiring a pre-alignment step. Moreover, the LoRA adapters may limit the ability of the model to shift its attention pattern.

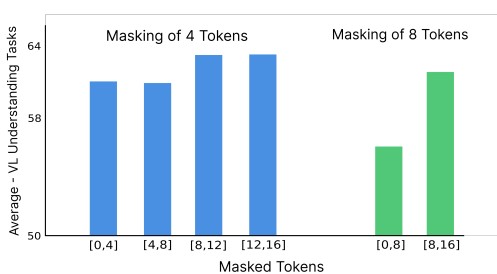

Figure 7: The relative importance of different subsets of visual tokens. We show the mean over the VL understanding tasks when masking specific subsets of the compressed visual tokens.

Table 18: Compression with Bidirectional vs Causal attention for generation (MMB, MME, TextVQA) and retrieval (Flickr30K, MS-COCO).

| Method | MMB | MME | TextVQA | Flickr30K | | MS-COCO | |
|---|---|---|---|---|---|---|---|
| | | | | T2I | I2T | T2I | I2T |
| Bidirectional | 60.2 | 1310.1 | 48.4 | 83.6 | 94.8 | 57.9 | 72.2 |
| Causal | 64.4 | 1470.0 | 54.2 | 83.8 | 94.4 | 56.8 | 70.2 |

**Finetuning checkpoint choice:** The natural starting point for our approach is the LLaVA model itself. However, for completeness, we also try to directly finetune from the Vicuna LLM itself. As the results from Table 19 show, starting from a model already optimized for vision-language understanding results in a notable performance boost. To compensate for this, likely, a longer training scheduler is needed and potentially a full model finetuning, as in LLaVA.

**Robustness to noisy inputs:** Following [5] we evaluate our approach under a various set of perturbations, e.g: zoom blur, elastic transformation, pixelation, JPEG compression, shot noise, brightness

Table 19: Impact of the pre-trained checkpoint for generation (MMB, MME, TextVQA) and retrieval (Flickr30K, MS-COCO).

| Method | MMB | MME | TextVQA | Flickr30K | | MS-COCO | |
|--------|-----|-----|---------|-----------|------|---------|------|
| | | | | T2I | I2T | T2I | I2T |
| Vicuna | 60.3 | 1296.3 | 48.2 | 81.2 | 92.5 | 54.3 | 67.4 |
| LLaVA | 64.4 | 1470.0 | 54.2 | 83.8 | 94.4 | 56.8 | 70.2 |

jitter, contrast jitter, Gaussian noise, etc. For brevity, we include in Table 20 the results in terms of relative performance drop on a subset of them. Notice that both approaches, with and without compression, have similar robustness strength.

**Latency measurements:** In the main manuscript we reported FLOPs estimates because the timings themselves are subject to the specific implementation and underlying hardware architecture. For completeness, we benchmark the LLaVA-1.5 7B model on a RTX 4090 GPU. Each result is averaged over 100 runs following a warm-up period. Original LLaVA model cost: 0.0587 sec/image (out of which 0.00353 sec spent for the vision encoder); Caching cost: 0.0584 sec/image; Online C&C cost (16 tokens): 0.00158 sec/image; Online C&C cost (4 tokens): 0.000406 sec/image; Caching cost + Online C&C (16 tokens): 0.0599 sec/image; Caching cost + Online C&C (4 tokens): 0.0588 sec/image. Our approach is significantly faster once the embeddings are cached and comparable with the LLaVA baseline during caching.

## C Additional details regarding the test-time inference

In Fig. 8, we depict the test-time inference flow for generative and discriminative tasks. The first step compresses the given image $\mathbf{X}_v$ into its compressed representation $\mathbf{H}_v^c$. This representation is then stored in a database. Note that while $\mathbf{H}_v^c$ can be computed on the fly, too, the scenario we are mostly interested in is pre-indexing, whereby the image representations are computed offline ahead of time.

Once stored, we can directly operate on this compressed representation for both generative and discriminative tasks. For generative tasks, the same LLM used for compression takes as input a user-provided instruction and the compressed image representation, producing an answer autoregressively (Fig. 8, top-right corner). For discriminative tasks, in order to measure the text-to-image similarity, á la CLIP, and again using the same LLM, we pass the user query (image description) to the LLM, producing a set of embeddings. We can measure the similarity between the given image description by taking the cosine similarity between the sum of the precomputed compressed vision tokens and the text embeddings newly produced by the LLM (Fig. 8, left side).

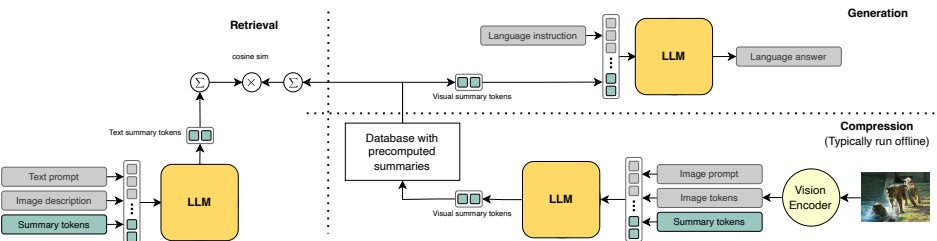

Figure 8: Test-time inference, depicting: compression (lower-right), generation (upper-right), and discrimination (left). Notice that in all cases we use the same LLM. The compressed embeddings are suitable for both sets of tasks and are generally expected to be pre-computed offline.

Table 20: Relative accuracy drop under various noise types across different datasets.

| Noise type | MMB | MME | POPE | SQA | TextVQA | realworldQA |
|---|---|---|---|---|---|---|
| Zoom Blur (baseline) | 20.45 | 0.0 | 0.0 | 7.31 | 0.0 | 17.15 |
| Zoom Blur (compressed) | 16.91 | 0.0 | 0.0 | 6.70 | 0.0 | 13.44 |
| Snow (baseline) | 11.04 | 0.0 | 0.0 | 2.51 | 0.0 | 7.73 |
| Snow (compressed) | 10.79 | 0.0 | 0.0 | 2.23 | 0.0 | 12.50 |
| Defocus Blur (baseline) | 12.50 | 0.0 | 0.0 | 3.17 | 0.0 | 7.49 |
| Defocus Blur (compressed) | 11.81 | 0.0 | 0.0 | 2.09 | 0.0 | 12.72 |
| Blank Image (baseline) | 73.38 | 42.21 | 43.42 | 9.52 | 90.14 | 22.95 |
| Blank Image (compressed) | 72.45 | 41.87 | 43.32 | 11.45 | 89.31 | 24.82 |
| Saturate (baseline) | 0.16 | 0.0 | 0.0 | 1.70 | 0.0 | 2.90 |
| Saturate (compressed) | 0.73 | 0.0 | 0.0 | 1.26 | 0.0 | 0.45 |
| Elastic Transform (baseline) | 5.52 | 0.0 | 0.0 | 2.36 | 0.0 | 3.62 |
| Elastic Transform (compressed) | 4.52 | 0.0 | 0.0 | 0.42 | 0.0 | 4.91 |
| Pixelate (baseline) | 8.44 | 1.99 | 9.00 | 0.89 | 68.08 | 13.77 |
| Pixelate (compressed) | 7.58 | 2.52 | 9.35 | 3.35 | 67.93 | 11.64 |
| Spatter (baseline) | 7.47 | 4.94 | 1.94 | 1.18 | 12.26 | 6.52 |
| Spatter (compressed) | 4.96 | 1.75 | 2.41 | 0.77 | 8.39 | 7.81 |
| Speckle Noise (baseline) | 10.88 | 3.01 | 3.29 | 2.36 | 15.24 | 8.21 |
| Speckle Noise (compressed) | 11.66 | 0.37 | 3.48 | 1.89 | 14.30 | 10.04 |
| JPEG Compression (baseline) | 2.60 | -0.89 | 2.24 | 0.0 | 5.68 | 4.83 |
| JPEG Compression (compressed) | 1.60 | 1.80 | 2.82 | -0.63 | 3.72 | 4.48 |
| Shot Noise (baseline) | 12.66 | 3.46 | 4.69 | 1.62 | 16.83 | 9.18 |
| Shot Noise (compressed) | 11.52 | 2.01 | 4.57 | 0.35 | 16.27 | 10.05 |
| Impulse Noise (baseline) | 12.01 | 4.64 | 4.35 | 1.99 | 16.30 | 8.21 |
| Impulse Noise (compressed) | 9.04 | 5.74 | 4.79 | 0.77 | 14.74 | 9.60 |
| Brightness (baseline) | 3.90 | 0.0 | 0.0 | 0.66 | 0.0 | 3.62 |
| Brightness (compressed) | 2.92 | 0.0 | 0.0 | -0.07 | 0.0 | -0.45 |
| Contrast (baseline) | 3.08 | 3.06 | 2.11 | 1.55 | 4.63 | 5.80 |
| Contrast (compressed) | 5.10 | 2.66 | 1.70 | 0.14 | 4.12 | 6.25 |
| Gaussian Noise (baseline) | 12.01 | 4.58 | 4.75 | 1.92 | 15.33 | 7.97 |
| Gaussian Noise (compressed) | 12.24 | 4.37 | 4.81 | 0.28 | 13.82 | 7.05 |
| Motion Blur (baseline) | 12.34 | 4.33 | 5.73 | 3.03 | 0.0 | 6.76 |
| Motion Blur (compressed) | 12.10 | 4.58 | 6.10 | 2.51 | 0.0 | 8.28 |

