# OpenReview forum: "Compress & Cache: Vision token compression for efficient generation and retrieval"
_NeurIPS.cc/2025/Conference — NeurIPS 2025 poster_

### Official Review · Reviewer_j93Q · 2025-06-21

**Clarity:** 3
**Significance:** 3
**Originality:** 3
**Rating:** 5
**Confidence:** 3

**Summary:**

The authors propose a method to optimize the inference speed of VLMs in a RAG scenario.

Specifically, given a pretrained VLM, the authors perform a first inference pass using an image, a pre-defined summarization prompt, and a small set of summary tokens. The summary tokens are then cached and used for further inferences alongside a query prompt.

The authors claim that this should significantly improve speed, as during "online" inference, the model would then process only around 6% of the image tokens, which the authors claim constitutes the overwhelming part of the input sequence compared to the textual query.

To make this possible, the authors fine-tune a pretrained VLM, in their case LLaVA 1.5, via LoRA adapters, using two losses:

1. A contrastive loss, applied on the summary tokens after the first forward pass.
2. An auto-regressive loss, applied on the generations produced by the model, given the text instructions and the summary tokens.

The two losses are then summed and jointly optimized using two LoRA adapters, one for summarization and one for generation. The adapters are trained for 10,000 steps using data from LLaVA-665K and CC3M, representing generative and contrastive tasks, respectively. The authors aim for a 1:1 sampling ratio between the two.

The fine-tuned models are then comprehensively evaluated across several text generation and image/text retrieval benchmarks, and the results show that C&C largely outperforms existing token pruning and matryoshka methods, even surpassing the original LLaVA baseline in several cases.

**Questions:**

1. How are the summary tokens initialized?
2. While the authors provide a FLOPs-based comparison between indexing and inference in Figure 5, I wonder how the inference speed would vary (and compare to other token pruning methods) under different scenarios, e.g., when the image is given by the user, and no retrieval is performed. Would it then be more efficient to use a matryoshka method compared to two VLM inference steps?
3. The paper states that they aim for a 1:1 sampling ratio between LLaVA 665k and CC3M. Do the authors aim for an overall 1:1 ratio, or do they strive to maintain this ratio in each batch? Would a different sampling ratio improve performance?
4. It would be interesting to know the total runtime of the LoRA fine-tuning process.
5. To my understanding, the first-stage LoRA adapters are optimized for both the contrastive and auto-regressive objectives, while the second-stage adapters are optimized only for the auto-regressive objective. Is this correct?

**Ethical Concerns:**

["NO or VERY MINOR ethics concerns only"]

**Final Justification:**

The authors properly addressed all my concerns. I have thus increased my score to accept.

**Limitations:**

The authors address some limitations of their work.

**Paper Formatting Concerns:**

N/A.

**Quality:**

3

**Strengths And Weaknesses:**

### Strengths

1. The paper is well-written and easy to follow.
2. The paper addresses a relevant problem, namely improving the inference speed of VLMs, with a focus on RAG/indexing scenarios, which are relevant for real-world use cases.
3. Their method is comprehensively evaluated across various datasets and tasks and compared to several recent token pruning and matryoshka methods, largely improving over them.

### Weaknesses

My primary concern regarding this paper is the confounding effect of LoRA fine-tuning. In particular, I am surprised to see that C&C improves over the baseline in several cases, e.g., +3.1% on VisWiz with 32 tokens, +0.1% on MMBench with only 16 tokens, representing a 97% reduction in the original number of tokens.

Thus, I wonder how much of this performance improvement (both over LLaVA and other methods) can be attributed to the LoRA fine-tuning and how much to C&C. I believe that a comparison between a LLaVA model fine-tuned via LoRA using a similar procedure (as much as possible) and C&C would provide clearer insights.

Beyond that, there are some minor details:

1. I believe that the dataset size comparison between EVA-CLIP and C&C on line 271 is unfair, as it compares the size of the dataset used for training this specific CLIP model versus the size of a fine-tuning dataset.
2. Table 1, for MMB, the best-performing method with 36 tokens is Matryoshka Multi, but its entry is not bolded.
3. Small typo in line 47 of the supplementary, "performs" should be "performance".

---

> ### Author Rebuttal · Authors · 2025-07-30
>
> We thank the reviewer for their time and feedback provided. We hope our responses below address their remaining concerns.
>
> **Q1.** _On confunding effects of LoRA fine-tuning C&C improving over the baseline in several cases. How much of this performance improvement (both over LLaVA and other methods) can be attributed to the LoRA fine-tuning and how much to C&C. I believe that a comparison between a LLaVA model fine-tuned via LoRA using a similar procedure (as much as possible) and C&C would provide clearer insights._
>
> **A1.** We believe the reasons behind the observed behaviours are multifold: (1) As shown in Table 1 in the paper, multiple compression methods report for certain datasets improvements over the uncompressed LLaVA-1.5 baseline (e.g. [28] on SQA and VisWiz, [3,28] on VisWiz, [3] on MMB). This suggests that the compressor acts as a filter that filters out some of the distractors, focusing more on prominent objects/concepts. (2) Performance on these datasets tends to vary between adjacent checkpoints during training, and the last checkpoint is not always the best one across all datasets, despite being the best on average. (3) The rest is indeed down to the LoRA finetuning, as the result of the experiment you suggested showcases below:
>
> | Compressor | GQA  | MMB  | MME    | POPE | SQA  | TextVQA | VisWiz |
> | ---------- | ---- | ---- | ------ | ---- | ---- | ------- | ------ |
> | LLaVA-1.5  | 62.0 | 64.3 | 1510.7 | 85.9 | 66.8 | 58.2    | 50.0   |
> | +LoRA      | 63.4 | 66.1 | 1496.9 | 86.4 | 68.4 | 58.2    | 49.8   |
>
> **Q2.** _I believe that the dataset size comparison between EVA-CLIP and C&C on line 271 is unfair, as it compares the size of the dataset used for training this specific CLIP model versus the size of a fine-tuning dataset._
>
> **A2.** What we meant is the size of the dataset used for _contrastive training_ (which is optimal for retrieval tasks). We will make this clear, thank you.
>
> **Q3.** _Table 1, for MMB, the best-performing method with 36 tokens is Matryoshka Multi, but its entry is not bolded. And a small typo in line 47 of the supplementary, "performs" should be "performance"._
>
> **A3.** Thank you! We have fixed them in the updated manuscript.
>
> **Q4.** _How are the summary tokens initialized?_
>
> **A4.** The summary tokens are initialized randomly. We explored multiple choice (e.g, zero-init, character-based init) but ultimately didn't notice a significant difference.
>
> **Q5.** _While the authors provide a FLOPs-based comparison between indexing and inference in Figure 5, I wonder how the inference speed would vary (and compare to other token pruning methods) under different scenarios, e.g., when the image is given by the user, and no retrieval is performed. Would it then be more efficient to use a matryoshka method compared to two VLM inference steps?_
>
> **A5.**  Following your suggestion, we benchmark the LLaVA-1.5 7B model on a RTX 4090 GPU. Each result is averaged over 100 runs following a warm-up period.
>
> Original LLaVA model cost: 0.0587 sec/image (out of which 0.00353 sec spent for the vision encoder)
>
> Caching cost: 0.0584 sec/image
>
> Online C&C cost (16 tokens): 0.00158 sec/image
>
> Matryoshka (16 tokens): 0.00521 sec/image
>
> Online C&C cost (4 tokens): 0.000406 sec/image
>
> Matryoshka (4 tokens): 0.004001 sec/image
>
> Caching cost + Online C&C (16 tokens): 0.0599 sec/image
>
> Caching cost + Online C&C (16 tokens): 0.0588 sec/image
>
> Notice that the practical measurements closely align with the expected approximated improvements. Moreover, in an online manner, once the features are cached, we are notably faster at building the context.
>
> Furthermore, our method can also make use of a sliced LLM to speed up the compression. To showcase this, we train a model with the following configuration: a LLaVA-OV 0.5B generator with a compressor instantiated using the first 25\% of the LLM's layer, as opposed to 100\%. This results in a 4x faster and smaller compressor. To recover the performance drop, we distill the features produced by the full-sized compressor. The results for 16 tokens (i.e. 45x compression rate, 729/16) are presented in the Table below:
>
> | Compressor | GQA | MMB | MME | POPE | SQA | TextVQA | VisWiz | realworldQA |
> |---|---|---|---|---|---|---|---|---|
> | none | 58.3 | 52.9 | 1461 | 88.3 | 67.2 | 65.8 | 47.4 | 54.1 |
> | 100\% (full) | 57.8 | 52.9 | 1516 | 88.2 | 70.9 | 64.3 | 48.9 | 54.5 |
> | 25\% | 57.1 | 49.2 | 1405 | 84.3 | 70.3 | 61.8 | 44.2 | 53.0 |
> | 25\% + distill | 58.0 | 51.1 | 1477 | 86.4 | 73.7 | 64.6 | 48.5 | 53.9 |
>
> This means that we can reduce the compression time by up to 4x without notable performance degradation.
>
> **Q6.** _The paper states that they aim for a 1:1 sampling ratio between LLaVA 665k and CC3M. Do the authors aim for an overall 1:1 ratio, or do they strive to maintain this ratio in each batch? Would a different sampling ratio improve performance?_
>
> **A6.** We aim for a 1:1 sampling ratio globally, not within a batch. The batches themselves are simply constructed to maximize throughput, grouping the samples by length. We did try a few different sampling ratios. The model is generally robust in the range 1:1 - 3:1 (665K:CC3M).
>
> **Q7.** _It would be interesting to know the total runtime of the LoRA fine-tuning process._
>
> **A7.** A training run took 40 hours on 24 GPUs.
>
> **Q8.** _To my understanding, the first-stage LoRA adapters are optimized for both the contrastive and autoregressive objectives, while the second-stage adapters are optimized only for the autoregressive objective. Is this correct?_
>
> **A8.** That's correct, we will make this clearer in the updated paper.

---

> > ### Comment · Reviewer_j93Q · 2025-08-03
> >
> > Thank you for your hard work preparing the rebuttal! You have properly addressed all my concerns. I will raise my score to accept.

---

> > > ### Author Response · Authors · 2025-08-03
> > > **Thank you for checking our rebuttal and suggestion**
> > >
> > > We thank the reviewer for checking our rebuttal and for the valuable suggestions made. We are happy to hear that our response has fully addressed your concerns.

---

### Official Review · Reviewer_h3Po · 2025-07-02

**Clarity:** 3
**Significance:** 3
**Originality:** 3
**Rating:** 5
**Confidence:** 3

**Summary:**

This paper presents a novel method, named C&C, to compress vision tokens into more compact features for a given VLM with minimal performance loss. The proposed method employs a “double-forward” calculation flow with two training objectives, enabling the VLM itself to extract compressed tokens that are simultaneously suitable for generation, discrimination, and storage. Extensive experiments across various scenarios demonstrate the effectiveness and efficiency of the proposed method.

**Questions:**

Regarding the storage perspective, the proposed method needs to store a summarized version of images with shape $R^{k' \times d}$, where $d$ is the hidden dimension of LLM. In contrast, the original LLaVA-1.5 only need to store features with shape $R^{k \times d_v}$, where $d_v < d$ is the dimension of the vision encoder output. It would be better to clarify this difference to illustrate the actual compression difference, rather than just comparing the number of tokens.

**Ethical Concerns:**

["NO or VERY MINOR ethics concerns only"]

**Final Justification:**

The authors' response has already resolved my concerns well. Since I had already expressed a clear intention to accept the paper, I will keep my score unchanged.

**Limitations:**

Yes.

**Paper Formatting Concerns:**

No.

**Quality:**

3

**Strengths And Weaknesses:**

### Strengths
1. The overall writing and organization of this paper is good.
2. The proposed method is simple, effective, and straightforward.
3. The experiments are sufficient to demonstrate the effectiveness of the proposed method.

### Weaknesses
The illustrations can be further refined for clarification and better readability. For example, Figure 2 does not show that the proposed method does not require finetuning the entire LLM, and the stage-specific LoRA modules are also missing from this figure.

---

> ### Author Rebuttal · Authors · 2025-07-30
>
> We thank the reviewer for their feedback and for recognizing the novelty of our approach. We hope our responses below address their remaining concerns.
>
> **Q1.** _The illustrations can be further refined for clarification and better readability. For example, Figure 2 does not show that the proposed method does not require finetuning the entire LLM, and the stage-specific LoRA modules are also missing from this figure._
>
> **A1.** Thank you for your suggestion. We have updated the figures and will include them in the updated manuscript.
>
> **Q2.** _Regarding the storage perspective, the proposed method needs to store a summarized version of images with shape $R^{k' \times d}$, where $d$ is the hidden dimension of LLM. In contrast, the original LLaVA-1.5 only need to store features with shape $R^{k \times d_v}$, where $d_v < d$ is the dimension of the vision encoder output. It would be better to clarify this difference to illustrate the actual compression difference, rather than just comparing the number of tokens._
>
> **A2.** You are generally right, if we opt to store the vision features prior to the projection layer that maps them to the input dimension of the LLM then generally $d_v < d$. The caveat is that we will have to rerun the projector in this case. For the LLaVA-1.5 7B model, $d=4096$ and $d_v=1024$ while for the smaller LLaVA-OV 0.5B model $d_v$ is a bit larger than $d$, with $d=896$ and $d_v=1152$.
>
> We will make this distinction clearer in the updated paper.

---

> > ### Comment · Reviewer_h3Po · 2025-08-05
> > **Reply to authors' response**
> >
> > Thank you to the authors for their response to my comments. Since I had already expressed a clear intention to accept the paper, I will keep my score unchanged.

---

> ### Comment · Area_Chair_SwUv · 2025-08-05
>
> Dear Reviewer h3Po,
>
> Please note that submitting mandatory acknowledgement without posting a single sentence to authors in discussions is not permitted. Whether your concerns have been addressed or not, please do tell the authors.
>
> Thanks,
>
> AC

---

### Official Review · Reviewer_LQY3 · 2025-07-02

**Clarity:** 3
**Significance:** 3
**Originality:** 3
**Rating:** 5
**Confidence:** 3

**Summary:**

This paper proposes Compress & Cache (C&C) to compressing vision tokens in Large Vision Language Models (LVLMs). Unlike prior works that focus on on-the-fly token compression, C&C introduces an offline token compression and caching strategy, effectively decoupling the compression process from inference. The author of the paper proposed double-forward pass training strategy, which uses the LVLM itself to generate compact visual summary tokens suitable for both generative and discriminative tasks. C&C employs both an autoregressive loss and a contrastive loss, and further improves performance using stage-specific LoRA adapters. The method achieves state-of-the-art results on a wide range of VQA, image captioning, and image-text retrieval benchmarks.

**Questions:**

please refer to weaknesses.

**Ethical Concerns:**

["NO or VERY MINOR ethics concerns only"]

**Final Justification:**

The rebuttal addresses my concerns. This paper demonstrates very solid works on vision token compression, which has potential for future research on VLMs. Therefore I suggest acceptance for this paper

**Limitations:**

yes

**Quality:**

3

**Strengths And Weaknesses:**

**Strengths**

1. The novelty of the paper is good. The paper introduces a double-forward pass strategy that reuses the same LVLM for both compression and generation.

2. The incorporation of both autoregressive loss (L_AR) and contrastive loss (L_disc) allows C&C to perform well in both domains.

3. The experiments performed in the paper is comprehensive. The author of the paper conducted sufficient tasks with C&C and compare with many baseline methods. The results shows robustness of the method.

4. The ablation and FLOPs results are very helpful.

**Weaknesses**

1. The most significant weakness of the proposed method is the lack of runtime latency results on the real hardware. The FLOPs analysis it good but it remains in theoretical computation. A comprehensive testing on GPU or other computing devices will be more convincing.

2. For any compression work, the main purpose is to maintain performance and reduce computation. Even the performance is good, it is still unknow if the robustness is affected. Some works report the result variance, or test the method in perturbed or adversarial input to show how robust the compressed model is. My concern on this paper is how the C&C performs when in such scenario?

---

> ### Author Rebuttal · Authors · 2025-07-30
>
> We thank the reviewer for their feedback and for recognizing the novelty of our approach. We hope our responses below address their remaining concerns.
>
> **Q1.** _The most significant weakness of the proposed method is the lack of runtime latency results on the real hardware. The FLOPs analysis it good but it remains in theoretical computation. A comprehensive testing on GPU or other computing devices will be more convincing._
>
> **A1.** We reported FLOPs estimates because the timings themselves are subject to the specific implementation. Following your suggestion, we benchmark the LLaVA-1.5 7B model on a RTX 4090 GPU. Each result is averaged over 100 runs following a warm-up period.
>
> Original LLaVA model cost: 0.0587 sec/image (out of which 0.00353 sec spent for the vision encoder)
>
> Caching cost: 0.0584 sec/image
>
> Online C&C cost (16 tokens): 0.00158 sec/image
>
> Online C&C cost (4 tokens): 0.000406 sec/image
>
> Notice that the practical measurements closely align with the expected approximated improvements: 576/16 = 36 theoretical vs 0.0587/0.00158 = 37.1,  respectively 576/4 = 144 theoretical vs 0.0587/0.000406 = 144.5. The practical gains are slightly higher because, in our case, the vision encoder doesn't need to be rerun.
>
> **Q2.** _For any compression work, the main purpose is to maintain performance and reduce computation. Even if the performance is good, it is still unknown if the robustness is affected. Some works report the result variance, or test the method in perturbed or adversarial input to show how robust the compressed model is. My concern on this paper is how the C&C performs when in such scenario?_
>
> **A2.** Thank you for your suggestion. Following [1] we evaluate our approach under a various set of perturbations, e.g: zoom blur, elastic transformation, pixelation, JPEG compression, shot noise, brightness jitter, contrast jitter, Gaussian noise, etc. For brevity, we include below the results in terms of relative performance drop on a subset of them. Notice that both approaches, with and without compression, have similar robustness strength.
>
>
> | Noise type | MMB | MME | POPE | SQA | TextVQA | realworldQA |
> | :--- | :--- | :--- | :--- | :--- | :--- | :--- |
> | **Zoom Blur (baseline)** | 20.45 | 0.0 | 0.0 | 7.31 | 0.0 | 17.15 |
> | **Zoom Blur (compressed)** | 16.91 | 0.0 | 0.0 | 6.70 | 0.0 | 13.44 |
> | **Snow (baseline)** | 11.04 | 0.0 | 0.0 | 2.51 | 0.0 | 7.73 |
> | **Snow (compressed)** | 10.79 | 0.0 | 0.0 | 2.23 | 0.0 | 12.50 |
> | **Defocus Blur (baseline)** | 12.50 | 0.0 | 0.0 | 3.17 | 0.0 | 7.49 |
> | **Defocus Blur (compressed)** | 11.81 | 0.0 | 0.0 | 2.09 | 0.0 | 12.72 |
> | **Blank Image (baseline)** | 73.38 | 42.21 | 43.42 | 9.52 | 90.14 | 22.95 |
> | **Blank Image (compressed)** | 72.45 | 41.87 | 43.32 | 11.45 | 89.31 | 24.82 |
> | **Saturate (baseline)** | 0.16 | 0.0 | 0.0 | 1.70 | 0.0 | 2.90 |
> | **Saturate (compressed)** | 0.73 | 0.0 | 0.0 | 1.26 | 0.0 | 0.45 |
> | **Elastic Transform (baseline)** | 5.52 | 0.0 | 0.0 | 2.36 | 0.0 | 3.62 |
> | **Elastic Transform (compressed)**| 4.52 | 0.0 | 0.0 | 0.42 | 0.0 | 4.91 |
> | **Pixelate (baseline)** | 8.44 | 1.99 | 9.00 | 0.89 | 68.08 | 13.77 |
> | **Pixelate (compressed)** | 7.58 | 2.52 | 9.35 | 3.35 | 67.93 | 11.64 |
> | **Spatter (baseline)** | 7.47 | 4.94 | 1.94 | 1.18 | 12.26 | 6.52 |
> | **Spatter (compressed)** | 4.96 | 1.75 | 2.41 | 0.77 | 8.39 | 7.81 |
> | **Speckle Noise (baseline)** | 10.88 | 3.01 | 3.29 | 2.36 | 15.24 | 8.21 |
> | **Speckle Noise (compressed)** | 11.66 | 0.37 | 3.48 | 1.89 | 14.30 | 10.04 |
> | **JPEG Compression (baseline)**| 2.60 | -0.89 | 2.24 | 0.0 | 5.68 | 4.83 |
> | **JPEG Compression (compressed)**| 1.60 | 1.80 | 2.82 | -0.63 | 3.72 | 4.48 |
> | **Shot Noise  (baseline)** | 12.66 | 3.46 | 4.69 | 1.62 | 16.83 | 9.18 |
> | **Shot Noise (compressed)** | 11.52 | 2.01 | 4.57 | 0.35 | 16.27 | 10.05 |
> | **Impulse Noise (baseline)** | 12.01 | 4.64 | 4.35 | 1.99 | 16.30 | 8.21 |
> | **Impulse Noise (compressed)** | 9.04 | 5.74 | 4.79 | 0.77 | 14.74 | 9.60 |
> | **Brightness (baseline)** | 3.90 | 0.0 | 0.0 | 0.66 | 0.0 | 3.62 |
> | **Brightness (compressed)** | 2.92 | 0.0 | 0.0 | -0.07 | 0.0 | -0.45 |
> | **Contrast (baseline)** | 3.08 | 3.06 | 2.11 | 1.55 | 4.63 | 5.80 |
> | **Contrast (compressed)** | 5.10 | 2.66 | 1.70 | 0.14 | 4.12 | 6.25 |
> | **Gaussian Noise (baseline)** | 12.01 | 4.58 | 4.75 | 1.92 | 15.33 | 7.97 |
> | **Gaussian Noise (compressed)**| 12.24 | 4.37 | 4.81 | 0.28 | 13.82 | 7.05 |
> | **Motion Blur (baseline)** | 12.34 | 4.33 | 5.73 | 3.03 | 0.0 | 6.76 |
> | **Motion Blur (compressed)** | 12.10 | 4.58 | 6.10 | 2.51 | 0.0 | 8.28 |
>
> (negative values denote cases where the performance increases marginally post-transformation).
>
> For completeness, we also report for a subset of transformations how the performance degradation changes under varying augmentation strengths (levels 3 and 5). The same conclusions hold true.
>
> | Noise type | MMB | MME | POPE | SQA | TextVQA | realworldQA |
> | :--- | :--- | :--- | :--- | :--- | :--- | :--- |
> | **Elastic Transform 5 (baseline)** | 18.34 | 0.0 | 0.0 | 3.10 | 0.0 | 10.39 |
> | **Elastic Transform 5 (compressed)**| 17.78 | 0.0 | 0.0 | 2.44 | 0.0 | 10.85 |
> | **Elastic Transform 3 (baseline)** | 5.52 | 0.0 | 0.0 | 2.36 | 0.0 | 3.62 |
> | **Elastic Transform 3 (compressed)**| 4.52 | 0.0 | 0.0 | 0.42 | 0.0 | 4.91 |
> | **Shot Noise 5 (baseline)** | 27.44 | 9.30 | 9.82 | 5.17 | 37.90 | 14.73 |
> | **Shot Noise 5 (compressed)** | 27.55 | 11.87 | 9.89 | 4.26 | 36.73 | 14.30 |
> | **Shot Noise 3 (baseline)** | 12.66 | 3.46 | 4.69 | 1.62 | 16.83 | 9.18 |
> | **Shot Noise 3 (compressed)** | 11.52 | 2.01 | 4.57 | 0.35 | 16.27 | 10.05 |
> | **Gaussian Noise 5 (baseline)** | 27.27 | 11.11 | 11.13 | 3.62 | 37.97 | 11.35 |
> | **Gaussian Noise 5 (compressed)**| 22.59 | 11.06 | 11.47 | 3.91 | 37.27 | 11.86 |
> | **Gaussian Noise (baseline)** | 12.01 | 4.58 | 4.75 | 1.92 | 15.33 | 7.97 |
> | **Gaussian Noise (compressed)**| 12.24 | 4.37 | 4.81 | 0.28 | 13.82 | 7.05 |
>
>
> [1] Benchmarking Robustness of Adaptation Methods on Pre-trained Vision-Language Models, Chen et al, NeurIPS 2023

---

> > ### Comment · Reviewer_LQY3 · 2025-08-03
> > **Thank you for the rebuttal**
> >
> > I appreciate the effort the author put for rebuttal. My concerns are fully addressed. I suggest the author to add hardware testing results in the final version of the paper. I will raise my score to accept.

---

> > > ### Author Response · Authors · 2025-08-03
> > > **Thank you for checking our rebuttal and your suggestions**
> > >
> > > We thank the reviewer for checking our rebuttal and for the valuable suggestions. We are happy to hear that our response has fully addressed their concerns, and we will certainly incorporate the suggested hardware testing results into the final manuscript.

---

### Official Review · Reviewer_jVQD · 2025-07-02

**Clarity:** 3
**Significance:** 3
**Originality:** 3
**Rating:** 4
**Confidence:** 4

**Summary:**

This paper introduces C&C (Compress & Cache), a novel paradigm for compressing vision tokens in Large Vision-Language Models (LVLMs). C&C decouples compression from inference by performing a one-time, offline indexing step where the LVLM itself compresses numerous vision tokens into a few compact "summary tokens." These cached tokens are then used for efficient online inference. The method is trained with a "double-forward pass" strategy, jointly optimized by an autoregressive loss for generation and a contrastive loss for discrimination. C&C aims to create a unified, nearly lossless representation for both generative and discriminative tasks, and experimental results show it sets a new state-of-the-art on various benchmarks, particularly in high-compression settings and for Visual RAG.

**Questions:**

n/a

**Ethical Concerns:**

["NO or VERY MINOR ethics concerns only"]

**Final Justification:**

I have reviewed the authors’ response as well as their discussion with the other reviewers. My concerns were addressed and I would like to maintain my recommendation to accept the paper.

**Quality:**

3

**Strengths And Weaknesses:**

**Strengths**
1. The core contribution is the "Compress & Cache" paradigm, shifting compression from an on-the-fly task to an offline process. This is a highly novel and practical approach, perfectly suited for real-world applications like Retrieval-Augmented Generation (RAG) and on-device deployment, where pre-indexing is feasible. It elegantly bypasses the inherent limitations of online compressors.

2. The "double-forward pass" strategy, which enables the LVLM to act as its own compressor, is a key technical highlight. This "self-compression" mechanism leverages the LLM's intrinsic capabilities for information integration, allowing it to create high-fidelity summary tokens without relying on external modules, which is crucial for its near-lossless performance.
Robust and Comprehensive Experiments: The paper's claims are substantiated by rigorous and comprehensive experiments. The extensive ablation studies (Tables 7, 8, 9) systematically validate each design choice, including the dual-loss function, the double-forward pass, and stage-specific LoRA adapters, providing strong evidence for the method's effectiveness.

**Weaknesses**
1. The method's core strength is also a limitation. It is not suitable for real-time applications (e.g., live video analysis) where pre-processing is impossible. While this is an intrinsic property of the paradigm, it constrains the method's scope of application.

2. The paper claims C&C is a general method, yet all experiments are confined to the LLaVA family. The LVLM landscape includes diverse architectures (e.g., Qwen-VL, CogVLM) with different vision-language interfaces. The effectiveness of the "self-compression" mechanism is not guaranteed to transfer across these architectures. By limiting validation to a single model family, the paper's claim of generalizability remains insufficiently supported by empirical evidence.

---

> ### Author Rebuttal · Authors · 2025-07-30
>
> We're grateful for your time and valuable feedback, and for recognizing the novelty of our approach! We hope our responses below address your remaining concerns.
>
> **Q1.** _The method's core strength is also a limitation. It is not suitable for real-time applications (e.g., live video analysis) where pre-processing is impossible. While this is an intrinsic property of the paradigm, it constrains the method's scope of application._
>
> **A1.** Our main focus is indeed on settings naturally suitable for offline indexing (e.g. RAG-based applications).
>
> However, we note that our method is sufficiently versatile, and the cost can be in part alleviated by combining our method with techniques like distillation and LLM slicing. To showcase this, we train a model with the following configuration: a LLaVA-OV 0.5B generator with a compressor instantiated using the first 25\% of the LLM's layer, as opposed to 100\%. This results in a 4x faster and smaller compressor. To recover the performance drop, we distill the features produced by the full-sized compressor. The results for 16 tokens (i.e. 45x compression rate, 729/16) are presented in the Table below:
>
> | Compressor | GQA | MMB | MME | POPE | SQA | TextVQA | VisWiz | realworldQA |
> |---|---|---|---|---|---|---|---|---|
> | none | 58.3 | 52.9 | 1461 | 88.3 | 67.2 | 65.8 | 47.4 | 54.1 |
> | 100\% (full) | 57.8 | 52.9 | 1516 | 88.2 | 70.9 | 64.3 | 48.9 | 54.5 |
> | 25\% | 57.1 | 49.2 | 1405 | 84.3 | 70.3 | 61.8 | 44.2 | 53.0 |
> | 25\% + distill | 58.0 | 51.1 | 1477 | 86.4 | 73.7 | 64.6 | 48.5 | 53.9 |
>
> **Q2.** _The paper claims C&C is a general method, yet all experiments are confined to the LLaVA family. The LVLM landscape includes diverse architectures (e.g., Qwen-VL, CogVLM) with different vision-language interfaces. The effectiveness of the "self-compression" mechanism is not guaranteed to transfer across these architectures. By limiting validation to a single model family, the paper's claim of generalizability remains insufficiently supported by empirical evidence._
>
> **A2.** Thank you for your suggestion. We would like to note that we have already validated the efficacy of our approach using two different architectures, LLaVA-1.5 and LLaVA-OneVision (LLaVA-OV). Although they partly share their name, they have different architectures and are trained on different data. Vision encoder: CLIP ViT-L/14 at 336px  (LLaVA-1.5) vs SigLIP ViT-SO400m/14 at 384px (LLaVA-OV); LLM: Vicuna-1.5 (LLaVA-1.5) vs Qwen2 (LLaVA-OV); Input: single patch and image at fixed 336px resolution (LLaVA-1.5) vs multi-patch (i.e. high resolution) 384-2304px operating range with multi-image support; Num vision tokens: 576 tokens per image (LLaVA-1.5) vs 729 tokens per patch (LLaVA-OV).
>
> In the main manuscript, the LLaVA-OV model was evaluated on the VisRAG suite. For consistency, in the table above, from answer A1, we report results on a shared suite of datasets. The same conclusions hold, our compressed models match and even outperform the original model in some cases.
>
> We note that many of the existing alternative models (e.g., QwenVL) are trained on closed-source data, making the retraining of the compressor under a fair setting impractical.

---

> > ### Comment · Reviewer_jVQD · 2025-08-04
> >
> > "We note that many of the existing alternative models (e.g., QwenVL) are trained on closed-source data, making the retraining of the compressor under a fair setting impractical."
> >
> > Does it mean it is large-scale data training needed for your method?

---

> > > ### Author Response · Authors · 2025-08-04
> > >
> > > Thank you for your question and for checking our rebuttal.
> > >
> > > It's not about the scale of the training data, but about having access to the same data that the original uncompressed model was trained on. All methods we compare with in Table 1 have the same requirement (i.e., access to the original training data).

---

> > > > ### Comment · Reviewer_jVQD · 2025-08-08
> > > >
> > > > My concerns are addressed.

---

### Comment · Area_Chair_SwUv · 2025-08-03

Dear Reviewers,

Thanks for your hard work during the review process. We are now in the author-reviewer discussion period.

Please (1) carefully read all other reviews and the author responses; (2) start discussion with authors if you still have concerns as early as possible so that authors could have enough time to response; (3) acknowledge and update your final rating. Your engagement in the period is crucial for ACs to make the final recommendation.

Thanks,

AC

---

### Decision · Program_Chairs · 2025-09-17

**Decision:**

Accept (poster)

**Comment:**

This paper presents a novel framework that compresses visual tokens in Large Vision-Language Models for both generative and discriminative tasks. Reviewers acknowledged the contribution and strong performance of the proposed method, while initially raising concerns regarding its efficiency and missing important experiments.

After the rebuttal, the authors addressed most of the concerns, and all reviewers agreed to accept this paper. AC read all the reviews, author rebuttals, and the paper, and believes this is a strong paper and recommends acceptance.